# Degradation of LMO2 in T cell leukaemia results in collateral breakdown of transcription complex partners and causes LMO2-dependent apoptosis

Naphannop Sereesongsaeng[1], Carole Bataille[2], Angela Russell[2,3], Nicolas Bery[4†], Fernando J Sialana[5], Jyoti Choudhary[5], Ami Miller[1,4‡], Terence Rabbitts[1,4*]

[1]Institute of Cancer Research, Division of Cancer Therapeutics, London, United Kingdom; [2]Department of Chemistry, Chemistry Research Laboratory, University of Oxford, Oxford, United Kingdom; [3]Department of Pharmacology, University of Oxford, Oxford, United Kingdom; [4]Weatherall Institute of Molecular Medicine, MRC Molecular Haematology Unit, University of Oxford, Oxford, United Kingdom; [5]Institute of Cancer Research, London, United Kingdom

*For correspondence:
terry.rabbitts@icr.ac.uk

Present address: †Université de Toulouse, Inserm, CNRS, Université Toulouse III-Paul Sabatier, Centre de Recherches en Cancérologie de Toulouse, Toulouse, France; ‡Evotec, Abingdon, United Kingdom

## eLife Assessment

This **important** paper reports the development of proteins and small molecules that induce degradation of a clinically-relevant oncogenic transcription factor, LMO2. The findings provide a proof of concept that PROTAC-type chemicals can be developed against intrinsically disordered proteins. The methods provide a blueprint for rational design of PROTACs starting from intracellular antibody paratopes. Overall, the paper is supported by **solid** evidence and will be of interest to chemical biologists and cancer pharmacologists.

**Abstract** LMO2 is an intrinsically disordered transcription factor activated in T cell leukaemia that is difficult to target. It forms part of a multi-protein complex that has bipartite DNA binding through heterodimeric basic-helix-loop-helix (bHLH) and GATA proteins. To determine if degradation of LMO2 in the context of T cell acute leukaemias (T-ALL) has therapeutic potential, a chimeric intracellular antibody has been developed fusing an anti-LMO2 single-domain variable region with one of three E3 ligases to create biodegraders. The intracellular binary interaction of these biodegraders with LMO2 leads to its proteasomal degradation but, in addition, concomitant loss of bHLH proteins that associate with LMO2 in the DNA-binding complex. Chemical compound surrogates of the intracellular antibody paratope (called antibody-derived [Abd] compounds) have been modified to create proteolysis targeting chimeras (PROTACs) for orthogonal assays of effects of LMO2 degradation. These form a ternary complex with LMO2 and E3 ligase in leukaemia cells that induces degradation of LMO2 and is also accompanied by loss of associated bHLH proteins. This is accompanied by T-ALL growth inhibition, alterations in proteins involved in cell cycling and instigation of apoptosis. These effects do not occur in the absence of LMO2. Our work demonstrates that degradation of LMO2 affects T-ALL, and the lead compounds can eventually be developed into drugs for patient treatment. Our work describes methods for drug discovery starting with antibody fragments.

## Introduction

Chromosomal translocations are among the consistent somatic mutations that lead to cancer and to the maintenance of the malignancy by abnormal gene activation and/or creating fusion genes (https://mitelmandatabase.isb-cgc.org/) (*Rowley, 2013*; *Rabbitts, 2009*). These changes are found in all tumour types, and regulation of the tumour state is due to the translocation gene expressed proteins. In tumours of haematopoietic origin and some solid tumours, for instance, sarcomas, the translocation proteins are transcription factors that can have master regulator functions in tumourigenesis by disrupting differentiation programmes (*Cleary, 1991*; *Rabbitts, 1991*). While translocation proteins offer tumour-specific targets and are therefore attractive as therapeutic targets, where they are transcription factors, they are considered undruggable, or hard-to-drug. This is because they are intrinsically disordered proteins or have extensive disordered regions that make them difficult to handle in recombinant form and for which limited structural information is known.

In T cell acute leukaemias (T-ALL), there are a number of chromosomal translocations (*Pagliaro et al., 2024*; *Williams et al., 1984*), including those activating the *LMO2* gene by t(11;14)(p13;q11) (*Boehm et al., 1991*; *Royer-Pokora et al., 1991*). The LMO2 protein is a master regulator of haematopoiesis since mouse gene targeting showed that both primitive (*Warren et al., 1994*) and definitive haematopoiesis fail with loss of the *Lmo2* gene (*Yamada et al., 1998*). Further, this regulator function of LMO2 is shown by its role in remodelling of established vascular endothelial but not in de novo endothelial formation (*Yamada et al., 2000*; *Yamada et al., 2022*). This latter function has implications for targeting tumour angiogenesis. LMO2 is a transcription factor whose role in creating a multi-protein complex was first demonstrated in erythroid cells, and the LMO2 complex comprises LDB1, TAL1, E47, and GATA (*Wadman et al., 1997*). Although LMO2 is a zinc-containing protein comprising two LIM domains, each with two LIM fingers, the protein does not appear to interact directly with DNA but rather bridges a multi-protein DNA-binding complex (*El Omari et al., 2013*).

LMO2 is an intrinsically disordered protein (IDP) that defied production in recombinant form and it has been hard to study the cellular properties of the protein and the protein complex. We have described anti-LMO2 human single variable intracellular domain antibody (designed iDAb) that blocks the formation of the LMO2 protein complex when expressed in cells (*Tanaka et al., 2011*) and discriminates the LMO family paralogues (*Sewell et al., 2014*). In addition, we used a cell-based intracellular antibody competition assay to develop compounds to interfere with the LMO2-iDAb interaction and these small molecules that are surrogates of the iDAb paratope (*Bery et al., 2021*) (antibody-derived [Abd] compounds). Further, chimeric intracellular antibodies could be engineered to carry warheads such as pro-caspase that cause proximity-induced, antigen-dependent apoptosis (*Tse and Rabbitts, 2000*) and suggested that other fusion proteins could be designed to affect other cellular pathways such as proteolysis (*Lobato and Rabbitts, 2003*). This was demonstrated using our anti-RAS iDAb for targeted protein degradation of KRAS (*Bery et al., 2020*). As a means to study the LMO2 transcription complex in the context of T-ALL, we have produced two types of protein degradation mediators. In the first, an anti-LMO2 iDAb was directly fused with one of three different E3 ligases (herein called biodegraders) and in the second, an LMO2 iDAb paratope-derived Abd compound was altered to be two different proteolysis targeting chimeras (PROTACs) (*Sakamoto et al., 2001*; *Békés et al., 2022*; *Paiva and Crews, 2019*; *Burslem and Crews, 2020*). Treatment of T-ALL cells expressing LMO2, from chromosomal translocations or other promoter alterations, with these degraders showed rapid targeted protein degradation of the LMO2 transcription factor but also levels of TAL1 and E47 (members of the transcription complex; *Wadman et al., 1997*) are reduced. These alterations in the LMO2 multi-protein complex cause inhibition of cell growth accompanied by initiation of programmed cell death. Therefore, targeting protein stability of the LMO2 complex shows the dependency of LMO2-positive T-ALL and that other proteins in the transcription factor complex can be disrupted by collateral degradation. LMO2 degraders are lead compounds for drug development for LMO2-positive T-ALL, and the approach is applicable to transcription factor complexes, including those involving IDPs.

## Results

### A biodegrader causes loss of LMO2 and collateral loss of TAL1/E47 bHLH

In our mouse models of LMO2 tumourigenesis, we observed that transgenic enforced thymus expression of Lmo2 (using CD2 promoter) but not Tal1 causes later onset, clonal T cell neoplasia, which arose earlier in double transgenic mice (*Larson et al., 1996*). Expressing anti-LMO2 iDAb (a human VH segment) inhibited tumour growth in a transplantation model (*Tanaka et al., 2011*). To assess the ability to degrade LMO2 by the ubiquitin proteasome system, we began to dissect the elements of interaction in the LMO2 complex by engineering chimeric intracellular antibodies where an E3 ligase was fused either at the N- or C-terminus of the iDAb to mediate binary interactions between iDAb-E3 ligase and LMO2 for ubiquitination of LMO2 and proteasomal degradation. Targeted protein degradation of LMO2 was observed when any of the six iDAb-E3 ligase proteins were expressed in HEK293T cells along with an LMO2-expressing plasmid. No detectable loss of control protein RAS was observed (or of the loading controls, cyclophilin-b, and β-actin). Conversely, using anti-RAS iDAb-CRBN or anti-KRAS DARPin-UBOX or anti-KRAS DARPin-VHL caused turnover of RAS but not LMO2 (*Figure 1—figure supplement 1*).

We further investigated the LMO2 degradation in natural LMO2 expression settings (i.e. T-ALL lines) using a lentivirus that conditionally expresses anti-LMO2 iDAb VH576 fused to CRBN E3 ligase (designated TLCV2-iDAb-CRBN) via a doxycycline-inducible promoter (TLCV2; *Barger et al., 2019*). CRBN was chosen in the inducible assay since thalidomide is commonly used in PROTAC chemistry, and this gave the opportunity to further evaluate LMO2 degradation by cereblon. This lentivirus was used to infect either Jurkat (lacking LMO2 expression), KOPT-K1 (*Dong et al., 1995*), or P12-Ichikawa cells (*Watanabe et al., 1978*) (expressing LMO2 after chromosomal translocations). The level of LMO2 expression after induction of the biodegrader was analysed by western blotting. LMO2 levels showed a marked decrease within 9 hr of iDAb-CRBN induction and continued for the 48 hr of analysis in LMO2-expressing cells (*Figure 1A*).

The LMO2 protein is associated with a bipartite DNA-binding multi-protein complex, involving basic-helix-loop-helix (bHLH) proteins TAL1/SCL and E47 and GATA (*Wadman et al., 1997*). We determined if other members of the complex could be degraded coincidentally with LMO2 using the anti-LMO2 biodegrader. KOPT-K1 cells, which express LMO2 from a t(11;14)(p13:q11) chromosomal translocation (*Dong et al., 1995*), were infected with the lentiviral TLCV2-iDAb-CRBN, and the biodegrader expression was induced for 9, 24, and 48 hr with doxycycline. Protein extracts were analysed by western blotting using antibodies recognising various components of the LMO2 complex. Targeted protein degradation of LMO2 is observed throughout the course of the biodegrader induction, and we also observed a rapid degradation of the two associated bHLH proteins TAL1/SCL and E47. No evidence was found for associated turnover of another bHLH protein LYL1 or of the LMO2-associating protein LDB1 or GATA3 (*Figure 1B*). This unexpected finding of the bHLH collateral degradation is unlikely to be due to off-target binding of the anti-LMO2 iDAb with TAL1 or E47.

### Control of LMO2 protein levels using chemical degraders

These results with the biodegraders demonstrate the degradability of endogenous LMO2 by E3 ligase-based molecules. The difficulty of the delivery of these macromolecules supported the subsequent development of small-molecule-based LMO2 degraders. To circumvent the problem of antibody delivery, we have previously developed a technology approach which uses antibody paratopes to select small-molecule surrogates (Abd technology) (*Quevedo et al., 2018*). Employing this technology, an Abd chemical series that binds to LMO2 was identified that interferes with the protein complex in cells (*Bery et al., 2021*). As a means to compare their cell-specific effects with the biodegraders, the Abd chemical compounds have been reformatted as PROTAC Abd degraders (*Li and Crews, 2022*). We used Abd-L21 and Abd-L27 as starting points for the PROTAC design (*Figure 2—figure supplement 1*) based on previously described SAR (*Bery et al., 2021*), where a broad tolerance of a range of para-substituents on the right-hand side arene ring was found (*Figure 2A*). Similarly, on the left-hand side, meta-substituted arenes were preferred with a range of substituents tolerated. Piperazine was preferred to the imidazolidin-2-one, seen in Abd-L9 (*Figure 2—figure supplement 1*), for activity, stability, and ease of synthesis, and was therefore selected for our PROTACs. Thiazole and oxazole were found to be interchangeable without any noticeable difference in activity, although

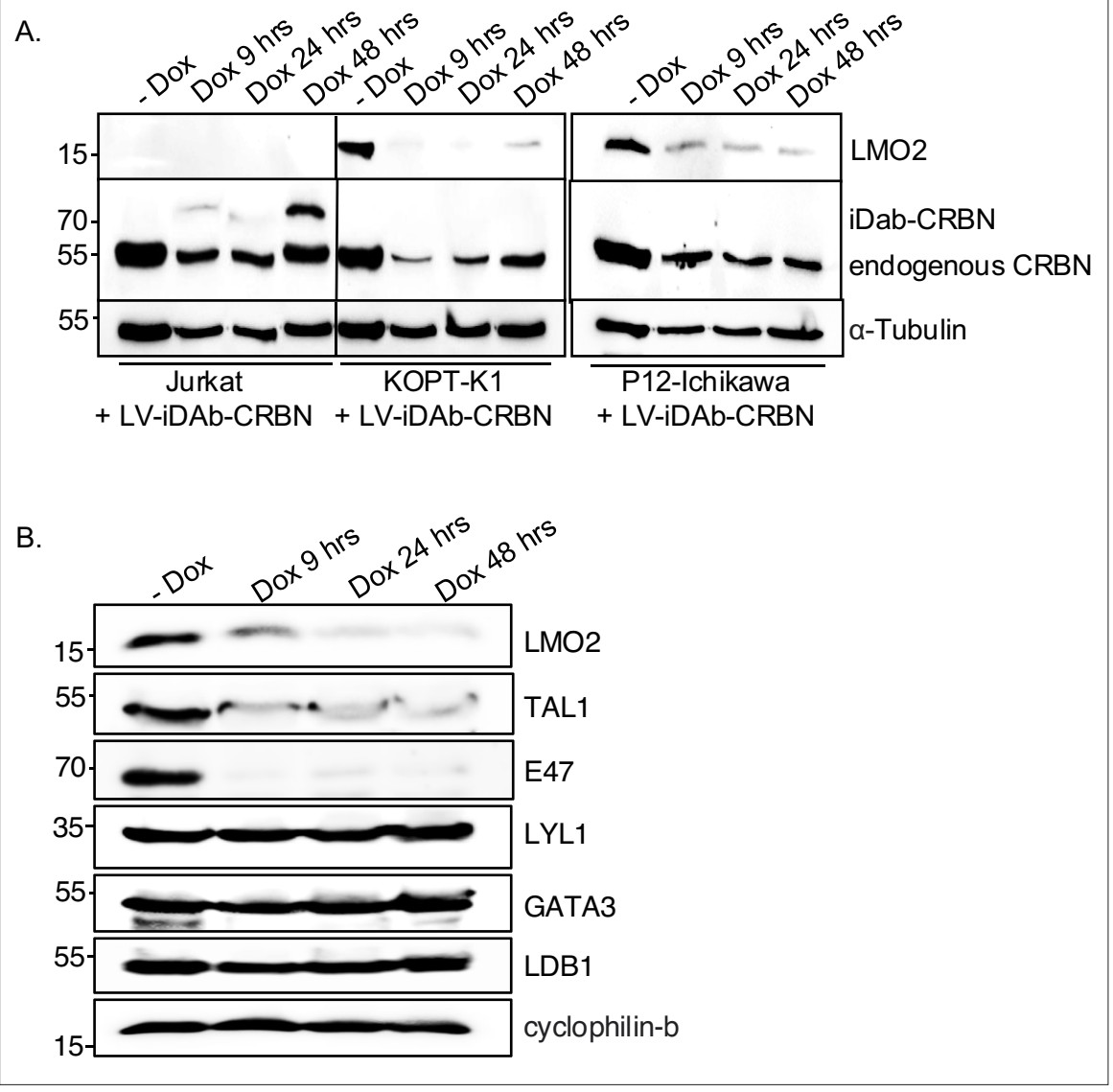

**Figure 1.** Lentivirally expressed biodegrader affects the LMO2 multi-protein complex in T cell lines. (**A**) The T cell lines Jurkat (LMO2-), KOPT-K1 (LMO2+), and P12-Ichikawa (LMO2+) cells were infected with lentivirus packaging plasmids and transfer vector (TLCV2-VH576-L10-CRBN) for 16 hr followed by 2 µg/ml doxycycline induction for 9, 24, and 48 hr. Western blotting analysis was used to detect LMO2, endogenous CRBN, and VH576-L10-CRBN levels after lentivirus infection. α-Tubulin was used as an internal loading control for western blotting analysis. (**B**) The level of LMO2 and proteins associated with the LMO2 transcription complex (TAL1, E47, Lyl-1, GATA3, and LDB1) was determined after KOPT-K1 cells were infected with TLCV2-VH576-L10-CRBN followed by 2 µg/ml doxycycline induction for 9, 24, and 48 hr. Cyclophilin-b was used as an internal loading control.

The online version of this article includes the following source data and figure supplement(s) for figure 1:

**Source data 1.** Western blot data with label shows the effect of lentiviral expressed biodegrader to the LMO2 multi-protein complex in T cell lines.

**Source data 2.** Western blot raw data shows the effect of lentiviral expressed biodegrader to the LMO2 multi-protein complex in T cell lines.

**Source data 3.** Western blot data with label shows the level of LMO2 and proteins associaed with the LMO2 transcription complex.

**Source data 4.** Western blot raw data shows the level of LMO2 and proteins associaed with the LMO2 transcription complex.

**Figure supplement 1.** LMO2 protein degradation in HEK293T cells after transfection with biodegrader constructs.

**Figure supplement 1—source data 1.** Western blot data with label shows LMO2 protein degradation in HEK293T cells with different biodegrader contruct.

**Figure supplement 1—source data 2.** Western blot raw data shows LMO2 protein degradation in HEK293T cells with different biodegrader contruct.

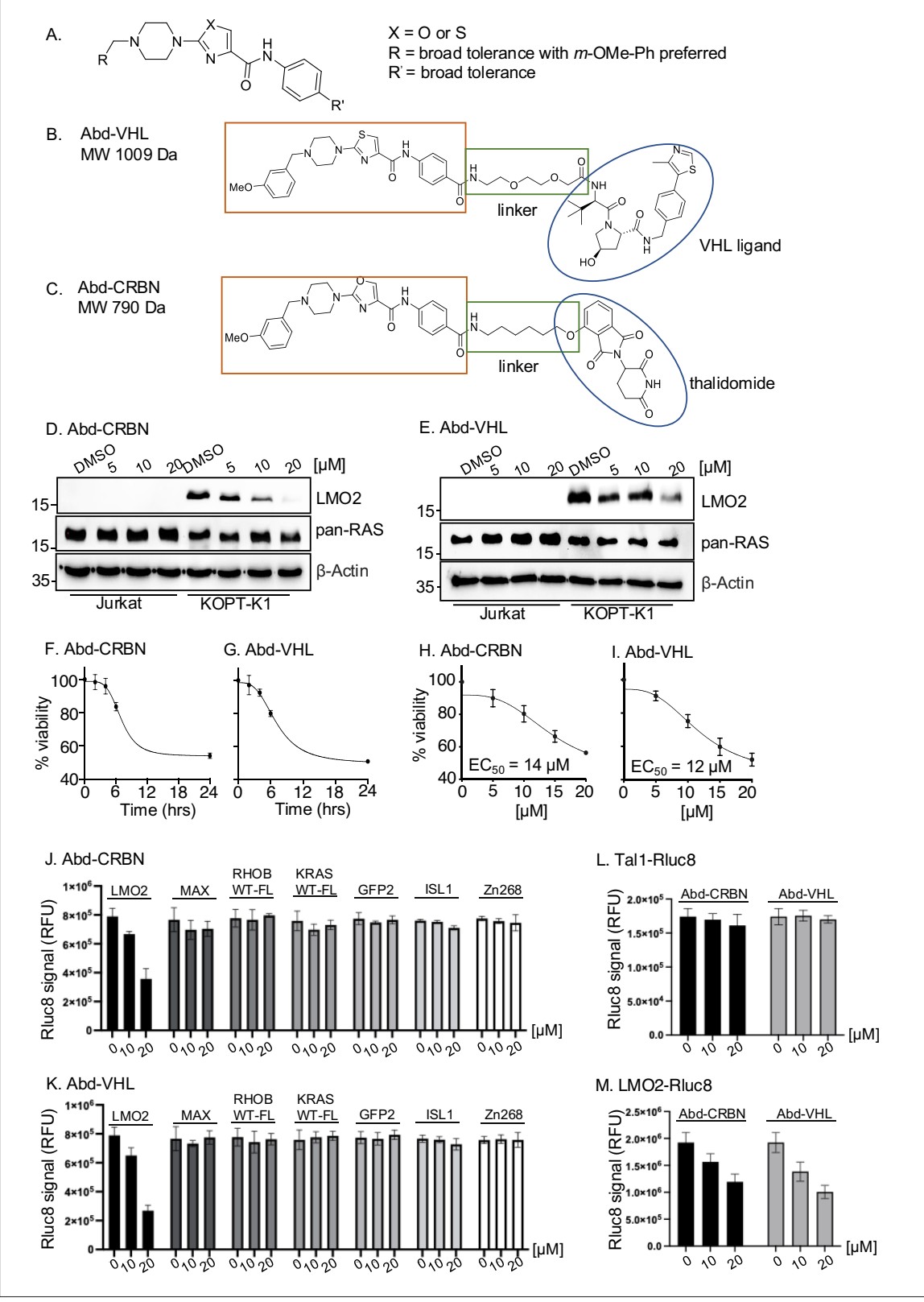

**Figure 2.** LMO2 antibody-derived (Abd) degraders and effects on cellular protein. (**A**) Structure-activity observations of LMO2-binding compounds derived using intracellular domain antibody (iDAb) VH576 in competitive, cell-based BRET (designated Abd compounds) contributed to a general structure of the LMO2-binding compounds. Based on broad tolerance to chemical modification at position R', two proteolysis targeting chimeras (PROTACs) were synthesised, one bearing the VHL ligand (Abd-VHL, **B**) and the other bearing thalidomide (the CRBN ligand) (Abd-CRBN, **C**). Each

*Figure 2 continued on next page*

*Figure 2 continued*

PROTAC thus comprises the LMO2-binding ligand, a linker and an E3 ligase ligand. (**D–E**) KOPT-K1 and Jurkat cells were treated with different concentrations of Abd degraders at 0, 5, 10, and 20 µM for 24 hr, protein extracts prepared and fractioned by SDS-PAGE followed by western blotting. Abd-CRBN was used on cells in (**D**) and Abd-VHL in (**E**). Cells treated with 1% DMSO were used as a control and β-actin was used as an internal loading control for the western blotting analysis. The viability of KOPT-K1 cells was treated with 15 µM Abd-CRBN (**F**) or 15 µM Abd-VHL (**G**) for 0, 2, 4, 6, and 24 hr, using CellTiter-Glo assay. Data are presented as a relative to luminescence at 0 hr, normalised to 100%. Determination of $DC_{50}$ values of Abd-CRBN (**H**) and Abd-VHL (**I**) after 24 hr treatment in KOPT-K1 was determined using CellTiter-Glo calculated by GraphPad Prism 9.0 software. All the values were presented as the average values relative to cell viability values in control (DMSO-treated cells) normalised to 100%. Data represent mean + SEM (n=3). Luciferase cell-based report assays were used to determine the degradation of the proteins after transfected HEK293T cells with different Renilla luciferase (Rluc8) reporter plasmids (pEF-LMO2-Rluc8, pEF-MAX-Rluc8, pEF-RHOBWT-FL, pEF-KRSWT-Fl-RLuc8, pEF-GFP2-Rluc8, pEF-ISL1-Rluc8, and pEF-Zn268) followed by Abd-CRBN (**J**) and Abd-VHL (**K**) treatment at 0, 10, and 20 µM for 24 hr. The potential degradation of pEF-Tal-Rluc8 (**L**) and pEF-LMO2-Rluc8 (**M**) was determined by luciferase cell-based report assays after the treatment with Abd-CRBN or Abd-VHL at 0, 10, and 20 µM for 24 hr. Data represent mean + SEM (n=3).

The online version of this article includes the following source data and figure supplement(s) for figure 2:

**Source data 1.** Raw data of cell viability and percentage of cell viability after Abd-CRBN treatment in KOPT-K1.

**Source data 2.** Raw data of cell viability and percentage of cell viability after Abd-VHL treatment in KOPT-K1.

**Source data 3.** Raw data of luminescence measurement after Abd-CRBN treatment in KOPT-K1 for 24 hr.

**Source data 4.** Raw data of luminescence measurement after Abd-VHL treatment in KOPT-K1 for 24 hr.

**Source data 5.** Raw data of luminescence measurement in reporter assay after Abd-CRBN treatment in KOPT-K1.

**Source data 6.** Raw data of luminescence measurement in reporter assay after Abd-VHL treatment in KOPT-K1.

**Source data 7.** Raw data of luminescence measurement of pEF-Tal1-Rluc8 in reporter assay.

**Source data 8.** Raw data of luminescence measurement of pEF-LMO2-Rluc8 in reporter assay.

**Figure supplement 1.** Viability of Jurkat and KOPT-K1 cells after treatment with antibody-derived (Abd) compounds and Abd degraders.

**Figure supplement 2.** LMO2 antibody-derived (Abd) degraders and effects on cellular proteins.

**Figure supplement 2—source data 1.** Western blot data with label shows LMO2 level in Jurkat and KOPT-K1 after the treatment from 0-20 µM at 2 and 6 hr.

**Figure supplement 2—source data 2.** Western blot raw data shows LMO2 level in Jurkat and KOPT-K1 after the treatment from 0-20 µM at 2 and 6 hr.

**Figure supplement 3.** Densitometry analysis of LMO2 protein expression.

**Figure supplement 4.** Dose response of T cell lines with antibody-derived (Abd) degraders.

**Figure supplement 5.** Longevity of responses of KOPT-K1 cells to single-dose treatment with antibody-derived (Abd) degraders.

**Figure supplement 5—source data 1.** Western blot data with label shows the longevity of responses of KOPT-K1 cells to single dose treatment.

**Figure supplement 5—source data 2.** Western blot raw data shows the longevity of responses of KOPT-K1 cells to single dose treatment.

**Figure supplement 6.** LMO2 levels and viability of KOPT-K1 cells treated with antibody-derived (Abd) degrader compounds and E3 ligase inhibitors.

**Figure supplement 6—source data 1.** Western blot data with label shows LMO2 level of KOPT-K1 treated with Abd compounds and inhibitors.

**Figure supplement 6—source data 2.** Western blot raw datashows LMO2 level of KOPT-K1 treated with Abd compounds and inhibitors.

the thiazole series was found to be easier to synthesise with higher chemical stability and better yields. It was decided to use a medium-length linker as a starting point, namely two PEG units for the VHL-binding ligand (*Buckley et al., 2012*) and a six-carbon alkyl unit for the CRBN-binding ligand (*Remillard et al., 2017*). Accordingly, two Abd E3 ligase ligand hetero-bifunctional molecules, designated Abd-VHL and Abd-CRBN, were synthesised with either the VHL or thalidomide E3 ligase ligand, respectively (*Figure 2B and C*).

The selectivity of the chemical degraders was tested by comparing T cell viabilities when treated with Abd-VHL and Abd-CRBN or the starting Abd compounds. The result showed that reformatting the LMO2 Abd compounds as PROTACs improved selectivity of LMO2-expressing cells compared to non-expressor ones. This difference was clearer when titrating the Abd-VHL and Abd-CRBN compounds over a concentration range during a 24 hr assay period. Jurkat (LMO2 negative) were largely unaffected, while KOPT-K1 (LMO2 positive) showed loss of viability with increased PROTAC concentration, which is an effect sustained over 48 hr (*Figure 2—figure supplement 1*).

The effect of the Abd-CRBN or Abd-VHL compounds on LMO2 protein levels was then assessed in dose response in KOPT-K1 and Jurkat cells. Cells were treated with a single dose of the compound, ranging from 0 to 20 µM for 2, 6, or 24 hr followed by analysis of protein levels by western blotting

analysis and cell viability. At 24 hr treatment, LMO2 levels are reduced by either Abd-CRBN (*Figure 2D*) or Abd-VHL (*Figure 2E*), whereas control protein RAS levels are unaffected. There is also a progressive loss of LMO2 protein from 6 hr onwards up to the end point of 24 hr (*Figure 2—figure supplement 2* and quantified by densitometry, *Figure 2—figure supplement 3*). These findings were paralleled with loss of cell viability over the 24 hr period when KOPT-K1 cells were treated with 15 µM of either compound (*Figure 2F–I*) showing the half-maximal effective concentration ($EC_{50}$) in KOPT-K1 was 14 and 12 µM for Abd-CRBN and Abd-VHL, respectively. The $EC_{50}$ values were in a comparable range to MOLT-3 cells (another LMO2-expressing cell line) compared to Jurkat cells that are resistant to either compound (*Figure 2—figure supplement 4*). The longevity of degradation was followed using western blotting and CellTitre-Glo assays (*Figure 2—figure supplement 5*). LMO2 was lost in KOPT-K1 during the 72 hr of cell culture after a single dose of compounds, while little or no effect was observed in the LMO2 non-expressing Jurkat cells range.

The possible effect of the compounds on non-LMO protein controls was assessed using a luciferase cell-based report assay (*Figure 2J and K*). Fusions with Renilla luciferase of various baits were expressed in HEK293T cells, including the LIM protein ISL1 and the zinc finger proteins Zn268, and luciferase levels determined after treatment with the PROTAC compounds (0–20 µM) for 24 hr. Only the loss of luciferase in the positive control LMO2-Rluc8 fusion was observed. In addition, confirmation that the specific involvement of the Abd-CRBN or Abd-VHL to the proteasome machinery in the LMO2 turnover was confirmed by using either proteasome inhibitors or competing the Abd degraders with the respective free E3 ligase ligands (*Figure 2—figure supplement 6*). Furthermore, TAL1 protein was fused to Renilla luciferase to investigate possible cross-reaction of the LMO2 PROTAC compounds with TAL1. After treatment with either of the PROTACs compounds (0–20 µM) for 24 hr, the luciferase level does not reduce in TAL1-Rluc8 (*Figure 2L*), whereas a reduction of luciferase signal in the positive control LMO2-Rluc8 fusion was observed (*Figure 2M*). This result suggests that the collateral degradation of TAL1/SCL and of E47 is due to biological effects beyond simple protein-compound interaction, as also found with the application of the biodegrader.

## Loss of LMO2 protein also affects partner proteins in the transcription complex

Lentiviral infection of KOPT-K1 cells expressing an LMO2 biodegrader showed collateral loss of TAL1 and E47 proteins with LMO2 (*Figure 1*). The bystander degradation was not evident for other members of the transcription complex. To gain further insights into this finding, the effect of the Abd-CRBN and Abd-VHL compounds was assayed with two LMO2-expressing cell lines (KOPT-K1 and CCRF-CEM) compared with two non-expressors (Jurkat and DND-41) (*Figure 3*). A similar pattern of LMO2 degradation was observed in the two expressing lines as we had first found with expressed biodegrader from the lentivirus. Allied to this, the loss of both TAL1 and E47 proteins is dose-dependent (*Figure 3A*), while there was no detectable loss of LYL1 (another T-ALL implicated bHLH protein), GATA3, or LDB1. Importantly, there was no apparent loss of these proteins in the two LMO2-non-expressing T cell lines. These results suggested that the proteolysis of TAL1 and E47 depended on degradation of LMO2 (*Figure 3B*). The loss of LMO2, TAL1, and E47 occurs rapidly as protein loss is observed in as little as 2–4 hr after treatment (*Figure 3C*). The involvement of the proteasome was confirmed by the inhibition of Abd-CRBN and Abd-VHL by epoxomicin in both KOPT-K1 and CCRF-CEM (*Figure 3D*) and requirement for E3 ligase by inhibition with the respective free ligand.

## The degradation of LMO2 interferes with cell growth, affecting proteins involved in proliferation

The potential effect of LMO2 protein loss was studied using the panel of cell lines expressing LMO2 (KOPT-K1, PF-382, P12-Ichikawa, CCRF-CEM, MOLT-16, and LOUCY) by simultaneously monitoring protein turnover and cell growth. Cells were treated with a single dose of either LMO2 PROTAC and analysed by western blot (*Figure 4A–F*). The western blot results showed that LMO2 degradation was observed to have occurred and was sustained for 48 hr. In this time frame, we did not find any evidence of loss of RAS proteins or β-actin controls.

Comparing the effects on growth of cells treated with Abd-CRBN or Abd-VHL showed an inhibitory effect on T cell growth if LMO2 is expressed (a significant reduction [p<0.001], *Figure 4G–L*), whereas T cells without LMO2 (Jurkat, DND-41, ALL-SIL, SUPT-1, and RPMI8402) are largely unaffected

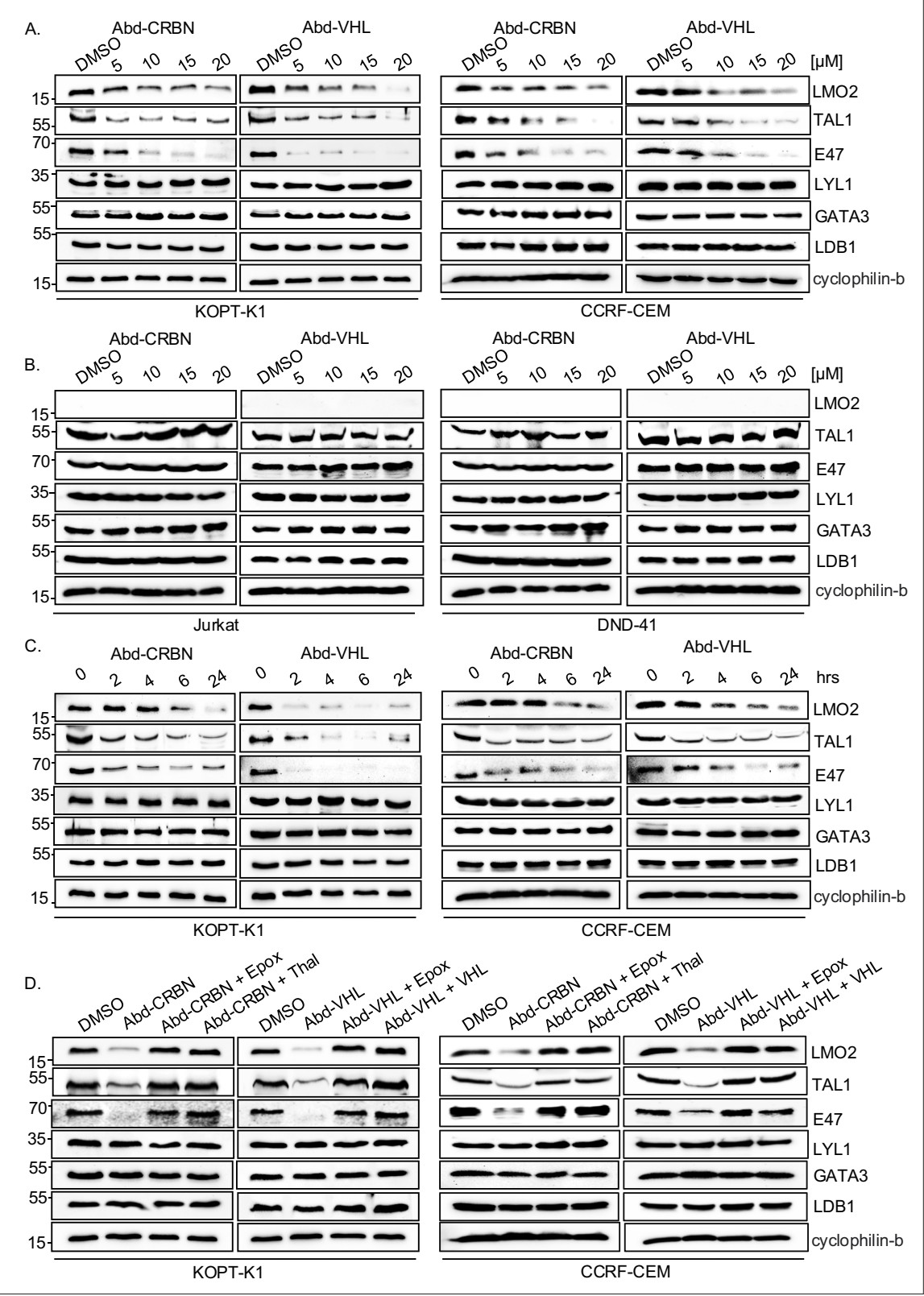

**Figure 3.** Proteins associated within the LMO2 transcription complex are co-degraded with the LMO2 proteolysis targeting chimeras (PROTACs). LMO2-expressing (KOPT-K1 and CCRF-CEM) and non-expressing T cells (Jurkat and DND-41) were treated with antibody-derived (Abd) degraders, and protein extracts were prepared for western blot, detecting LMO2, TAL1, E47, Lyl-1, GATA3, LDB1 proteins (cyclophilin-b was used as an internal loading control) after cells were treated with different concentrations of Abd degraders at 0, 5, 10, 15, and 20 µM for 24 hr. (**A**) Western blotting analysis with

*Figure 3 continued on next page*

*Figure 3 continued*

KOPT-K1 and CCRF-CEM extracts and (**B**) western blotting analysis with Jurkat and DND-41 extracts. (**C**) KOPT-K1 and CCRF-CEM were treated with Abd degraders at 20 µM for 2, 4, 6, and 24 hr. Western blotting analysis data show LMO2, TAL1, and E47 proteins expression in KOPT-K1 and CCRF-CEM that was affected by treatment. (**D**) The involvement of proteasome machinery in protein complex degradation was investigated in KOPT-K1 and CCRF-CEM with either proteasome inhibitors or by competing the potential of the Abd degraders with free E3 ligase ligand. Western blot data show LMO2, TAL1, and E47 expression in KOPT-K1 and CCRF-CEM after the treatment with or without inhibitors followed by Abd compounds. Inhibitors used were the proteasome inhibitor epoxomicin (0.8 µM), or CRBN inhibitors (10 µM thalidomide) or free VHL ligand. Cyclophilin-b was used as an internal loading control for western blotting analysis.

The online version of this article includes the following source data and figure supplement(s) for figure 3:

**Source data 1.** Western blot data with label shows ptoteins associated within the LMO2 transcription complex in KOPT-K1 and CCRF-CEM.

**Source data 2.** Western blot raw data shows ptoteins associated within the LMO2 transcription complex in KOPT-K1 and CCRF-CEM.

**Source data 3.** Western blot data with label shows ptoteins associated within the LMO2 transcription complex in Jurkat and DND-41.

**Source data 4.** Western blot raw data shows ptoteins associated within the LMO2 transcription complex in Jurkat and DND-41.

**Source data 5.** Western blot data with label shows ptoteins associated within the LMO2 transcription complex in KOPTK1- and CCRF-CEM after treated with Abd-degraders at 20 µM.

**Source data 6.** Western blot raw data shows ptoteins associated within the LMO2 transcription complex in KOPTK1- and CCRF-CEM after treated with Abd-degraders at 20 µM.

**Source data 7.** Western blot data with label shows the involvement of proteasome machinery in protein complex degradation.

**Source data 8.** Western blot raw data shows the involvement of proteasome machinery in protein complex degradation.

**Figure supplement 1.** Half-lives of LMO2 and LMO2 protein complex in KOPT-K1 cells.

**Figure supplement 1—source data 1.** Western blot data with label shows half-lives of LMO2 and LMO2 protein complex in KOPT-K1.

**Figure supplement 1—source data 2.** Western blot raw data shows half-lives of LMO2 and LMO2 protein complex in KOPT-K1.

**Figure supplement 2.** The LMO2 transcription complex indicating lysine residues in the basic-helix-loop-helix (bHLH) heterodimer for potential lysine ubiquitination.

(*Figure 4M–Q*). The absence of LMO2 protein was confirmed by western blotting, RT-PCR analysis (*Figure 4—figure supplements 1–2*), and genomic sequencing (*Figure 4—figure supplement 3*), and those expressing LMO1 were confirmed by RT-PCR. RPMI8402 has an LMO1 chromosomal translocation t(11;14)(p15;q11) from which *LMO1* was cloned (*Boehm et al., 1988*; *McGuire et al., 1989*). The small effects observed at 48 hr may be attributable to off-target toxicity.

Given the short time of exposure to the Abd PROTAC compounds (24 hr) to impair LMO2-positive T-ALL cell growth, we performed proteomic analysis of cells treated with Abd-VHL to ascertain the resultant changes to the cell composition. Proteomic data were obtained in an analysis of Abd-VHL treated in KOPT-K1 (LMO2 expressor), Jurkat (non-expressor) and RPMI8402 (non-expressor) cells. After the treatment of 15 µM Abd-VHL for 24 hr, the effect on the level of LMO2 was determined using liquid chromatography-mass spectrometry (LC-MS) with a targeted MS2 method (*Figure 5A*) and western blotting (*Figure 5B*), which confirmed the significant reduction of LMO2 after the treatment in KOPT-K1. We collected total proteome dataset and conducted Gene Ontology (GO) and KEGG pathway analyses using rank-based Gene Set Enrichment Analysis (GSEA) across the three cell lines and visualised the resulting enrichment scores in a heatmap (*Figure 5C*). Several pathways are affected after the treatment with Abd-VHL, even in the two cell lines that do not express LMO2. Most importantly, proteins related to cell division appear in the top 10 significantly affected pathways in KOPT-K1, while this effect does not appear in Jurkat and RPMI8402, consistent with cell growth data in KOPT-K1. Further, the average fold change (log$_2$FC) in KOPT-K1 (*Figure 5D*) and Jurkat (*Figure 5E*) was represented as volcano plots, indicating that the total significantly changed proteins in KOPT-K1 is higher than Jurkat cells. To specify the effect of the Abd-VHL treatment to proteins changed in KOPT-K1, the proteins significantly increased or decreased compared to Jurkat cells are presented in *Figure 5D* and tabulated in *Figure 5F* (and in *Figure 5—source data 3*). The result showed that those significantly down-regulated proteins in KOPT-K1 relate to cell division pathway and some aspects of ubiquitination, while most proteins in Jurkat cells are unchanged. Overall, these results help explain why treatment with Abd-VHL has an effect on cell division in the LMO2-expressing cells.

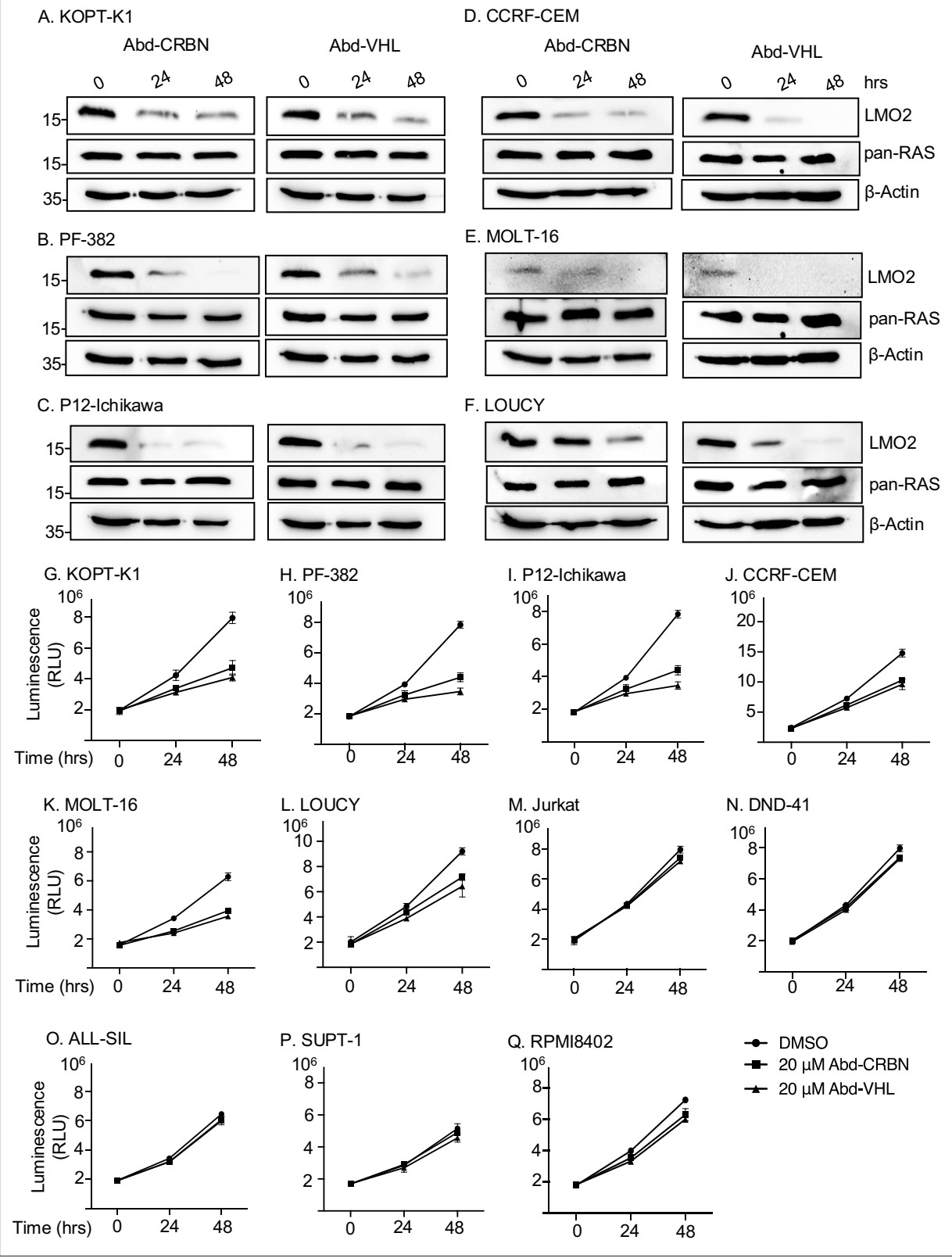

**Figure 4.** Loss of LMO2 protein inhibits T cell acute leukaemia (T-ALL) cell growth. LMO2-expressing T cells were treated with 20 μM antibody-derived (Abd)-CRBN or Abd-VHL for 24 and 48 hr, and protein extracts were prepared for western blot to detect LMO2 protein, RAS protein as negative control, and β-actin was used as an internal loading control. Western blotting analysis data show LMO2 expression in KOPT-K1 (**A**), PF-382 (**B**), P12-Ichikawa (**C**), CCRF-CEM (**D**), MOLT-16 (**E**), and LOUCY (**F**) that was affected by treatment with the LMO2 proteolysis targeting chimera (PROTAC) compounds.

*Figure 4 continued on next page*

*Figure 4 continued*

Comparative cell numbers were determined from increase in viability determined using the CellTiter-Glo assay. Data represent mean + SEM (n=3). LMO2-expressing cell lines tested were KOPT-K1 (**G**, t(11;14)(p13;q11)), PF-382 (**H**), P12-Ichikawa (**I**, t(11;14)(p13;q11)), CCRF-CEM (**J**), MOLT-16 (**K**), and LOUCY (**L**). LMO2 non-expressing T cells were also treated with 20 µM Abd-CRBN or Abd-VHL for 24 and 48 hr after which relative luminescence was determined using the CellTiter-Glo assay. Cells tested were Jurkat (**M**), DND-41 (**N**), ALL-SIL (**O**), SUPT-1 (**P**), and RPMI8402 (**Q**). Data represent mean + SEM (n=3).

The online version of this article includes the following source data and figure supplement(s) for figure 4:

**Source data 1.** Western blot data with label shows the loss of LMO2 expression in T-ALL cells.

**Source data 2.** Western blot raw data shows the loss of LMO2 expression in T-ALL cells.

**Source data 3.** Raw data of luminescence measurement from CellTiter-Glo assay in T-ALL cells after the treatment of Abd-CRBN and Abd-VHL for 24 and 48 hr.

**Figure supplement 1.** Verification of LMO2 non-expressing T cell lines.

**Figure supplement 1—source data 1.** Western blot data with label shows the verification of LMO2 non-expressing T-cells.

**Figure supplement 1—source data 2.** Western blot raw data shows the verification of LMO2 non-expressing T-cells.

**Figure supplement 2.** Reverse transcription-PCR (RT-PCR) analysis of the human T cell acute leukaemia (ALL) panel.

**Figure supplement 2—source data 1.** Agarose gel data with label shows PCR products from RT-PCR analysis of the human T-ALL cells.

**Figure supplement 2—source data 2.** Agarose gel raw data shows PCR products from RT-PCR analysis of the human T-ALL cells.

**Figure supplement 3.** Confirmation of chromosomal translocation in KOPT-K1 tissue culture cells.

**Figure supplement 3—source data 1.** Agarose gel data with label shows PCR products to confirm the chromosomal translocation in KOPT-K1.

**Figure supplement 3—source data 2.** Agarose gel raw data shows PCR products to confirm the chromosomal translocation in KOPT-K1.

## Loss of the LMO2 protein complex initiates apoptosis

We investigated whether LMO2 protein degradation in the T-ALL cell lines resulted in programmed cell death that accompanies growth inhibition effects. KOPT-K1 or Jurkat cells were treated with a single 20 µM dose of either Abd-CRBN or Abd-VHL for up to 30 hr and caspase 3/7 activation determined at the various time points. Increased levels of caspases were found in LMO2-expressing KOPT-K1 treated with either Abd compound, rising 10-fold (from 23,325 RLU at 4 hr to 228,865 RLU at 30 hr) using Abd-CRBN and 25-fold increase (from 19,460 RLU at 4 hr to 476,843 RLU at 30 hr) using Abd-VHL (*Figure 6A*). This rise in activated caspases mirrors loss of LMO2 as judged by western blotting analysis (*Figure 6B*) and with appearance of cleaved caspase 3, cleaved caspase 7, and cleaved poly(ADP-ribose) polymerase (PARP). The increase of caspase 3/7 activity in KOPT-K1 cells treated with Abd-CRBN is lower than observed with Abd-VHL, reflected by relatively low levels of protein detected in western blot.

Conversely, the effect of both compounds on the LMO2-negative Jurkat cells was a small increase in caspase 3/7 levels over the time course of the experiment. This increased 2.6-fold (from 16,223 RLU at 4 hr to 43,651 RLU at 30 hr) using Abd-CRBN and 4.5-fold (from 20,309 RLU at 4 hr to 92,683 RLU at 30 hr) using Abd-VHL. Western blotting analysis up to 48 hr of treatment with compounds did not show significant levels of activated, cleaved caspase 3, and only small amounts of activated, cleaved caspase 7 at 48 hr after treatment with Abd-CRBN (*Figure 6B*). Similarly, Abd-VHL treatment (*Figure 6C*) of Jurkat cells did not induce detectable cleaved caspase 3 or 7 in the western blotting analysis. Small amounts of cleaved PARP were detected in the Western blotting analysis for treatment with both Abd compounds. Cells treated with only DMSO in culture medium showed essentially no detectable activated caspase 3/7 during this growth period (*Figure 6A*) nor activated, cleaved caspase or PARP (*Figure 6B and C*). To corroborate the detection system for cleaved caspase 7, both Jurkat and KOPT-K1 cells were treated with the chemotherapy reagent doxorubicin and analysed by western blotting and luminescence assays (*Figure 6—figure supplement 1*).

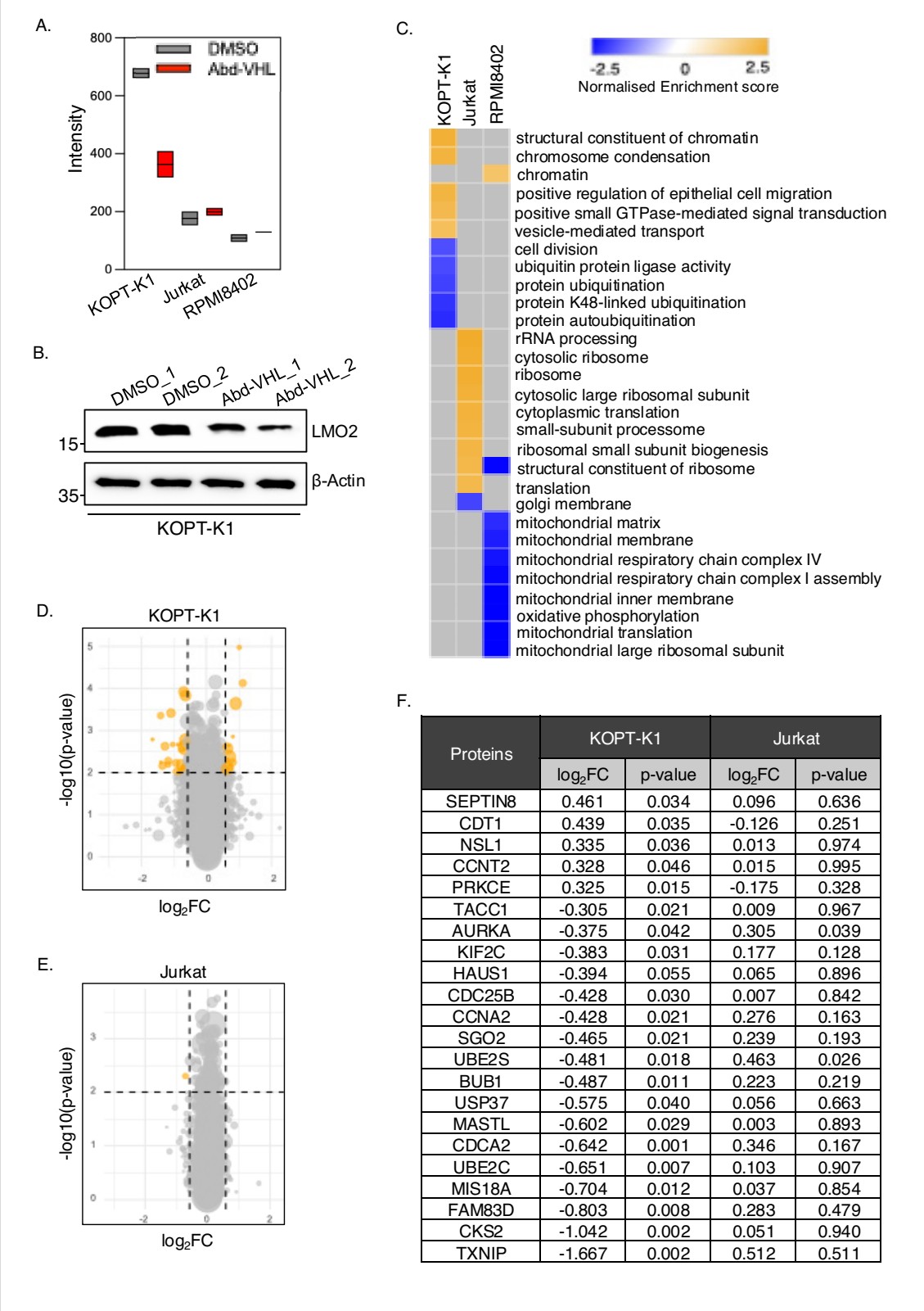

**Figure 5.** Proteomic profiling of T cell acute leukaemia (T-ALL) cells after treatment with antibody-derived (Abd) proteolysis targeting chimera (PROTAC) degrader. LMO2-expressing T cells (KOPT-K1) and LMO2 non-expressing T cells (Jurkat and RPMI8402) were treated in duplicate with 15 μM Abd-VHL for 24 hrs. (**A**) Floating bar plots of LMO2 protein levels measured by targeted proteomics in control (DMSO treated) (grey bars) and Abd-VHL treated (red bars) in KOPT-K1, Jurkat, and RPMI8402 cells. (**B**) Western blotting analysis of LMO2 levels in KOPT-K1 cells when treated with DMSO only and Abd-

*Figure 5 continued on next page*

*Figure 5 continued*

VHL for 24 hr. β-Actin protein detection was used as an internal loading control for western blotting analysis. (**C**) Heatmap displaying enriched terms from Gene Set Enrichment Analysis (GSEA) in KOPT-K1, Jurkat, and RPMI8402 cells after Abd-VHL treatment compared to control treatment (DMSO), based on global proteomics analysis. The enrichment score ranges from –2.5 (blue) to 2.5 (yellow). (**D and E**) Volcano plot of proteomic changes following Abd-VHL treatment compared to control treatment (DMSO) in KOPT-K1 (**D**) and Jurkat (**E**). The plots are based on the fold change ($\log_2$FC) and p-value ($-\log_{10}$(p-value)). Respectively, the yellow or grey circles indicate proteins with statistically significant or non-significant up-regulation/down-regulation. The cut-off for the volcano plots is $\log_2$FC = 0.585, p-value = 0.05. (**F**) A tabulation of statistically significant proteins in KOPT-KI, represented as yellow circles (**D**) compared to Jurkat cells. The data are presented as fold change ($\log_2$FC) and p-value ($-\log_{10}$(p-value)). The source data are available as *Figure 5—source data 3*.

The online version of this article includes the following source data for figure 5:

**Source data 1.** Western blot data with label shows LMO2 level in KOPT-K1 after treated with DMSO and Abd-VHL for 24 hr.

**Source data 2.** Western blot raw data shows LMO2 level in KOPT-K1 after treated with DMSO and Abd-VHL for 24 hr.

**Source data 3.** Proteins reduced by treatment of KOPT-K1 cells with Abd-VHL.

## Discussion

### LMO2 depletion by antibody biodegraders or PROTACs interferes with T-ALL

Transcription factors like LMO2 have been considered as hard-to-drug targets because they are involved in protein-protein or protein-DNA interaction and often have large intrinsically disordered regions, as in the case of LMO2 (*Sewell et al., 2014*). Therefore, using intracellular antibodies as inhibitors is a convenient starting point for drug discovery with these IDPs. In this study, we have shown that protein degradation of the LMO2 transcription factor occurs after either a binary interaction of an iDAb-E3 ligase fusion with LMO2 or a ternary interaction through PROTACs. Using an intracellular antibody for study of IDPs offers unique options because of the natural property of antibodies of high discriminating power, they can be easily manipulated by protein engineering (*Winter, 1989*) and can be used for target validation as a step before embarking on small-molecule drug development campaigns.

The function of LMO2 to regulate cell proliferation has been found involved during tumourigenesis, and the inhibition of LMO2 showed apoptosis induction in breast and colorectal cancer (*Liu et al., 2016*). In acute myeloid leukaemia cells, the LMO2/LDB1 complex has an important role to promote cell growth and proliferation and is required for cell survival (*Lu et al., 2023*). Our data show that the depletion of LMO2 has an effect on growth of T-ALL-expressing LMO2 but that the compounds have no off-target effect on non-expressing LMO2 T-ALL cells. Proteomic analysis (*Figure 5*) exploring the pathways associated with LMO2 protein reduction in T-ALL cells supports the conclusion that LMO2 down-regulation has an effect on proteins required for cell division, coupled to down-regulation of components of the ubiquitination machinery, known to coordinate phases and progression of cell division (*Zou and Lin, 2021*). Suppression of ubiquitination reduces proliferation and induces apoptosis, which was observed in treated cells (*Figure 6*). Further, several proteins involved in mitosis are reduced upon LMO2 degradation (annotated in *Figure 5—source data 3*), such as CCNA2 (cyclin A2), which controls G1/S and the G2/M transitions (*Zou and Lin, 2021*), is reduced in the LMO2 PROTAC-treated cells. In relation to ubiquitination itself and protein turnover, it is significant that both UBE2C and UBE2S E2 ubiquitin enzymes are reduced in LMO2-depleted cells. These two E2 enzymes ubiquitinate anaphase-promoting complex/cyclosome required proteasomal loss of cyclin B1 for exiting from mitosis (*Alfieri et al., 2017*). Overall, the data suggest that when LMO2 is depleted from the T-ALL cells, the effect is to block mitosis and apoptosis follows.

### Collateral turnover of transcription factor partners with LMO2 proteasome degradation

The LMO2 protein is a transcription factor that operates as a bridging molecule for a multi-protein complex that binds to DNA through bipartite regions comprising a bHLH heterodimer of TAL1 and E47 (binding to E box elements) and a GATA protein (binding to GATA elements) (*Wadman et al., 1997*). While LIM-only proteins have two LIM domains, each with two LIM zinc-binding fingers, these proteins do not interact directly with DNA. The LMO2 protein complex is required for function within T-ALL, and forced expression of LMO2 and TAL1 in transgenic mouse models demonstrated this

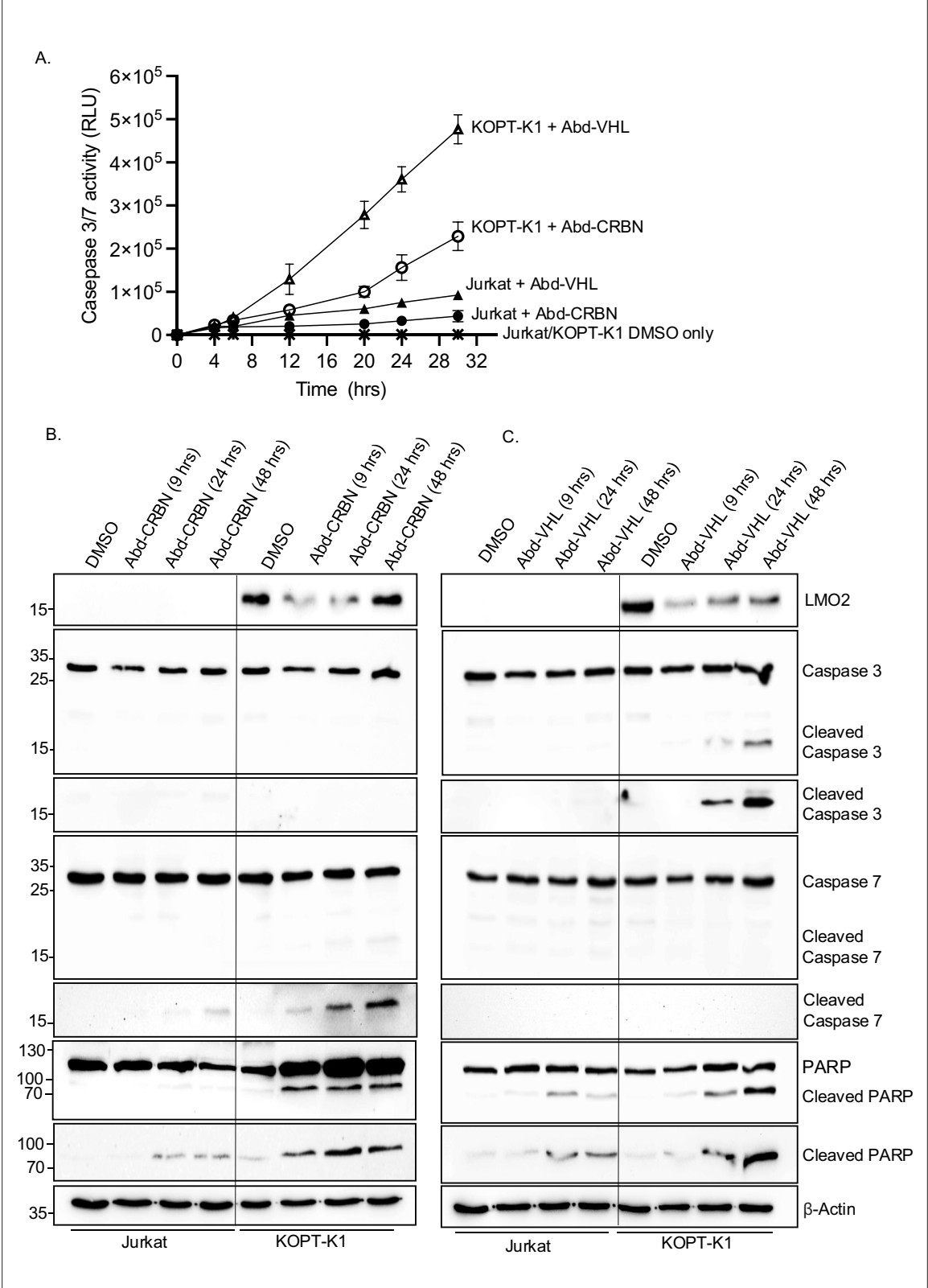

**Figure 6.** Caspase and poly(ADP-ribose) polymerase (PARP) cleavage in cells after treatment with antibody-derived (Abd) degraders indicative of apoptosis initiation. KOPT-K1 and Jurkat cells were treated with Abd-CRBN or Abd-VHL and effects on viability and programmed cell death were analysed at different times. (**A**) Time course of progressive expression of caspases after the treatment with compounds was assayed for caspase 3/7 levels using Caspase-Glo 3/7. Cells were treated with 20 µM Abd-CRBN or Abd-VHL followed by cell culture up to 30 hr. Cells treated with 1% DMSO

*Figure 6 continued on next page*

*Figure 6 continued*

in culture medium were used as a control. Data represent mean + SEM (n=3). KOPT-K1 and Jurkat cells were treated with 20 µM Abd-CRBN (**B**) or Abd-VHL (**C**) and cultured for 24 or 48 hr. Cell extracts were made and proteins subjected to western blotting analysis with specific antibodies for detection of LMO2, caspase 3, cleaved caspase 3, caspase 7, cleaved caspase 7, PARP, and cleaved PARP. β-Actin protein detection was used as an internal loading control for western blotting analysis.

The online version of this article includes the following source data and figure supplement(s) for figure 6:

**Source data 1.** Western blot data with label shows the caspase and PARP cleavge in Jurkat and KOPT-K1 after treated with Abd-CRBN or Abd-VHL for 24 and 48 hr.

**Source data 2.** Western blot raw data shows the caspase and PARP cleavge in Jurkat and KOPT-K1 after treated with Abd-CRBN or Abd-VHL for 24 and 48 hr.

**Source data 3.** Raw data of caspase3/7 level after the treatment with 20 µM Abd-CRBN or Abd-VHL.

**Figure supplement 1.** Caspase 7 and cleaved caspase 7 in Jurkat and KOPT-K1 cells after the treatment of doxorubicin.

**Figure supplement 1—source data 1.** Western blot data with label shows the expression of caspase 7 and cleaved caspase 7 in Jurkat and KOPT-K1 after the treatment of doxorubicin.

**Figure supplement 1—source data 2.** Western blot raw data shows the expression of caspase 7 and cleaved caspase 7 in Jurkat and KOPT-K1 after the treatment of doxorubicin.

leukaemogenesis cooperativity (*Larson et al., 1996*). However, in the experimental system described here, we have demonstrated the effects on cell growth by treating T cells with the LMO2 PROTACs, and both TAL1 and E47 were rapidly and simultaneously degraded with LMO2 (*Figures 1 and 3*). Since there is a close association of the two bHLH with LMO2 proteins within the protein complex, lysines in these two proteins that lie close to the PROTAC-binding site (*Figure 3—figure supplement 2*) could also undergo ubiquitination. Such an effect was described as bystander ubiquitination (*Bond and Crews, 2021*). Alternatively, degradation of LMO2 could destabilise the whole complex, in a natural process of transcription complex regulation, causing degradation of TAL1 and E47, as previously reported with PROTACs targeting the BAF chromatin remodelling complex (*Farnaby et al., 2019*). In addition, the half-lives of proteins in the LMO2 protein complex were examined using cycloheximide to block protein synthesis. The result showed that TAL1 and E47 have a short half-life (10 min and 1 hr, respectively, in this assay, *Figure 3—figure supplement 1*) and LMO2 protein has a half-life of around 6 hr (*Layer et al., 2020*).

We could not find evidence of differential loss of other members of the complex (LDB1 and GATA) or LYL1, which is a different bHLH protein implicated in T-ALL. This may be because bystander ubiquitination could not occur due to the interaction surfaces of these proteins in relation to the PROTAC docking sites. Alternatively, the disparity in the degradation of LMO2, TAL1, and E47 compared with LDB1 and GATA proteins may be attributed to different pool sizes for LMO2-bound and free components of the transcription complex. The loss of TAL1 and E47 bHLH proteins along with LMO2 in the degradation cascade is, however, a complementary effect that enhances the effectiveness of anti-LMO2 PROTAC compounds.

## A strategy for drug discovery against IDPs

Either of the two compounds described here may be developed for specifically treating T-ALL expressing LMO2. Furthermore, our data also illustrate a strategy for drug discovery for IDPs, estimating that up to 30% cellular proteins have large intrinsically disordered regions (*Deiana et al., 2019*), for instance, transcription factors. Starting with a domain antibody that had been used for LMO2 target validation (*Tanaka et al., 2011*), compound surrogates were discovered using the antibody as a screening tool (*Bery et al., 2021*) and the compounds developed using medicinal chemistry. This is a general approach for hard-to-drug, disordered proteins such as transcription factors.

## Materials and methods

**Key resources table**

| Reagent type (species) or resource | Designation | Source or reference | Identifiers | Additional information |
|---|---|---|---|---|
| Antibody | Anti-LMO2 (Goat polyclonal) | R&D Systems | Cat#AF2726 RRID:AB_2249968 | WB (1:1000) |
| Antibody | Anti-pan-RAS (Mouse monoclonal) | Sigma-Aldrich | Cat# OP40-100UG RRID:AB_10683383 | WB (1:1000) |
| Antibody | Anti-CRBN (Rabbit monoclonal) | Cell Signaling | Cat# 71810, RRID:AB_2799810 | WB (1:1000) |
| Antibody | Anti-VHL (Rabbit monoclonal) | Thermo Fisher Scientific | Cat# PA5-27322, RRID:AB_2544798 | WB (1:1000) |
| Antibody | Anti-PARP (Rabbit polyclonal) | Cell Signaling | Cat# 9542, RRID:AB_2160739 | WB (1:1000) |
| Antibody | Anti-TAL1 (Rabbit polyclonal) | A gift from Prof. Richard Bear (Columbia University) | N/A | WB (1:500) |
| Antibody | Anti-E2A (E47) (Rabbit polyclonal) | Cell Signaling | Cat# 4865, RRID:AB_10560512 | WB (1:1000) |
| Antibody | Anti-GATA-3 (Rabbit monoclonal) | Cell Signaling | Cat# 5852, RRID:AB_10835690 | WB (1:5000) |
| Antibody | Anti-LYL1 (Rabbit polyclonal) | Thermo Fisher Scientific | Cat# PA5-68772, RRID:AB_2691101 | WB (1:5000) |
| Antibody | Anti-LDB1 (Rabbit polyclonal) | Abcam | Cat# ab96799, RRID:AB_10679400 | WB (1:5000) |
| Antibody | Anti-caspase-3 (Rabbit monoclonal) | Cell Signaling | Cat# 9665, RRID:AB_2069872 | WB (1:1000) |
| Antibody | anti-cleaved caspase-3 (Rabbit monoclonal) | Cell Signaling | Cat# 9664, RRID:AB_2070042 | WB (1:1000) |
| Antibody | Anti-caspase-7 (Rabbit polyclonal) | Cell Signaling | Cat# 9492, RRID:AB_2228313 | WB (1:1000) |
| Antibody | Anti-cleaved caspase-7 (Rabbit polyclonal) | Cell Signaling | Cat# 9491, RRID:AB_2068144 | WB (1:1000) |
| Antibody | Anti-FLAG M2 (Mouse monoclonal) | Sigma-Aldrich | Cat# F1804, RRID:AB_262044 | WB (1:10,000) |
| Antibody | Anti-β-Actin (Mouse antibody) | Sigma-Aldrich | Cat# A5441, RRID:AB_476744 | WB (1:10,000) |
| Antibody | Anti-cyclophilin-B (Rabbit polyclonal) | Abcam | Cat# ab178397, RRID:AB_2924975 | WB (1:10,000) |
| Antibody | Anti-α-Tubulin (Rabbit polyclonal) | Abcam | Cat# ab4074, RRID:AB_2288001 | WB (1:10,000) |
| Antibody | Anti-goat IgG HRP-linked antibody (Rabbit polyclonal) | Abcam | Cat# ab6741, RRID:AB_955424 | WB (1:10,000) |
| Antibody | Anti-rabbit IgG HRP-linked antibody | Cell Signaling | Cat# 7071, RRID:AB_2099234 | WB (1:10,000) |
| Antibody | Anti-mouse IgG HRP-linked antibody (Horse polyclonal) | Cell Signaling | Cat# 7076, RRID:AB_330924 | WB (1:10,000) |
| Commercial assay or kit | CellTiter-Glo | Promega | Cat# G8090 | |
| Commercial assay or kit | Caspase-Glo 3/7 | Promega | Cat# G7570 | |
| Commercial assay or kit | Pierce BCA Protein assay kit | Thermo Scientific | Cat# 23225 | |
| Chemical compound, drug | von Hippel-Lindau (VHL) Ligand 1 | Cayman Chemical | Cat# 21591 | |
| Chemical compound, drug | Cereblon (CRBN) | A gift from Dr. Habib Bouguenina | N/A | |

| Reagent type (species) or resource | Designation | Source or reference | Identifiers | Additional information |
|---|---|---|---|---|
| Chemical compound, drug | Thalidomide | A gift from Dr. Habib Bouguenina | N/A | |
| Chemical compound, drug | Lenalidomide | A gift from Dr. Habib Bouguenina | N/A | |
| Chemical compound, drug | Epoxomicin | A gift from Dr. Habib Bouguenina | N/A | |
| Chemical compound, drug | MLN4924 | R&D Systems | Cat# 6499 | |
| Software, algorithm | GraphPad Prism | GraphPad Prism | RRID:SCR_002798 | |

## Molecular cloning

The iDAb LMO2-E3 ligase fusions were generated by polymerase chain reaction (PCR) using Phusion High-Fidelity PCR kit (NEB). pEF-VH576-L10-FLAG-CRBN-mb was used as a template to generate VH576-L10-FLAG-CRBN; pEF-FLAG-CRBN-L5-VH576-myc was used as a template to generate FLAG-CRBN-L5-VH576; pEF-VHL-L5-FLAG-VH576-mb was used as a template to generate VHL-L5-FLAG-VH576; pEF-NtFarn-FLAG-VH576-L10-VHL was used as a template to generate FLAG-VH576-L10-VHL; pEF-UBOX-L5-FLAG-VH576-mb was used as a template to generate UBOX-L5-FLAG-VH576; pEF-NtFarn-FLAG-VH576-L5-UBOX was used as a template to generate FLAG-VH576-L5-UBOX. All the PCR templates were previously described (*Bery et al., 2021*). PCR products were cloned into pEF-BOS plasmid and sequenced to confirm insert with the vector.

Generation of LMO2, MAX, RHOBWT-FL, KRASWT-FL, GFP$^2$, ISL-1, and Zn268 was by PCR using Phusion High-Fidelity PCR kit (NEB) and cloned into pEF-BOS-Rluc8 plasmid to produce pEF-LMO2-Rluc8, pEF-MAX-Rluc8, pEF-Rluc8-RHOBWT-FL, pEF-Rluc8-KRASWT-FL, pEF-GFP$^2$-Rluc8, pEF-ISL-1-Rluc8, and pEF-Zn268-Rluc8 (*Bery et al., 2021*; *Bery and Rabbitts, 2022*).

## Cell culture

Human T-ALL cell lines – KOPT-K1, Jurkat, MOLT-3, ALL-SIL, DND-41, PF382, SUPT-1, RPMI8402, and P12-Ichikawa – were cultured in RPMI-1640 medium (Gibco) supplemented with 10% foetal bovine serum (FBS) (Pan-Biotech). HEK293T cell line was cultured in DMEM (Gibco) supplemented with 10% FBS. Cells were grown at 37°C in a humidified incubator with 5% $CO_2$ and were regularly performed mycoplasma tests using a MycoAlert Mycoplasma Detection Kit (Lonza). Cells that were mycoplasma-free were used for experiments.

## Transient transfection

HEK293T cells were seeded on a six-well plate at $3\times10^5$ cells/well and incubated at 37°C with 5% $CO_2$ until 70–80% confluence determined visually. After 24 hr, cells were transfected with Lipofectamine 2000 reagent (Invitrogen) according to the manufacturer's instructions. An iDAb LMO2-E3 ligase plasmid (ranging from 0 to 1.0 µg) was mixed with 1 µg pEF-BOS-LMO2 expression vector (*Osada et al., 1995*) and diluted in 150 µl Opti-MEM reduced serum medium (Gibco). 10 µl Lipofectamine 2000 was diluted in 150 µl Opti-MEM reduced serum medium. The diluted DNA was added to diluted Lipofectamine 2000 and incubated for 15 min at room temperature. After the incubation, the mixture was added to the HEK293T cells. After 24 hr of transfection, the transfected cells were detached from the plate using trypsin-EDTA (0.05%), Phenol Red (Gibco) and incubated at 37°C for 2 min. Two volumes of pre-warmed complete media were added to inactivate trypsin, and dispersal of the medium, by pipetting over the cell layer surface, was used to recover >95% of cells. Cell pellets were collected by centrifugation at 100×*g* for 5 min at room temperature.

## Luciferase reporter assay

HEK293T cells were transiently transfected with 500 ng reporter vectors (pEF-LMO2-Rluc8, pEF-MAX-Rluc8, pEF-RHOBWT-FL-Rluc8, pEF-KRASWT-FL-Rluc8, pEF-GFP$^2$-Rluc8, pEF-ISL-1-Rluc8, or pEF-Zn268-Rluc8). After 24 hr, the transfected cells were re-seeded in white opaque 96-well plate at $5\times10^4$ cells/well and incubated at 37°C for 4 hr prior to the compound addition. The treated cells were incubated at 37°C for 24 hr, after which 10 µl of 250 µM luciferase substrate (Coelenterazine 400a) (Cayman Chemical) were added per well to get 10 µM at final concentration. The luminescence signal was measured at 400–700 nm wavelength filter luminescence using PHERAstar FSX (BMG Labtech).

## Lentiviral cell lines

The lentiviral transfer vector plasmid TLCV2-VH576-L10-FLAG-CRBN was constructed with TLCV2 (*Barger et al., 2019*). The pMA-VH576-L10-FLAG-CRBN was synthesised from GeneArt (Thermo Fisher). The insert was removed from the pMA vector by digesting with AgeI and NheI, then cloned into TLCV2 vector between Age/NheI sites to produce a TLCV2-VH576-L10-FLAG-CRBN plasmid.

Lentiviral particles were produced by transient co-transfection of lentiviral transfer vector plasmid and the packaging plasmids (pRSV-Rev, pMDLg/pRRE, and pMD2-VSV-G) in HEK293T cells using Lipofectamine 2000. After 48 hr transfection, debris was removed by spinning at 300×$g$ for 5 min at 4°C, followed by filtration of the supernatants through 0.45 μm filter to remove viral aggregates. The filtered lentiviral supernatants were centrifuged at 15,000×$g$ for 24 hr at 4°C. Virus pellets were resuspended in completed media and stored at –80°C in cryotubes. The lentiviral particles were titrated by serial dilution (3× dilution) in complete media. After 48 hr post-transduction, cells were harvested and analysed by flow cytometry for the viral GFP reporter expression and the viral titres calculated.

## T cell transduction

KOPT-K1 and Jurkat cells were infected by using TransDux MAX virus transduction reagent and followed the spinoculation method (SBI, System Bioscience). For the spinoculation, KOPT-K1 and Jurkat were seeded in a 24-well plate at 5×$10^5$ cells/well and resuspended with 400 μl complete RPMI media plus 100 μl MAX Enhancer, 2 μl TransDux, and 4 μl 1 M HEPES pH 7.0 buffer. Virus suspensions were added to cells at a multiplicity of infection of 3, and the plate was centrifuged at 1500×$g$ at 32°C for 2 hr. After spinoculation, the presence of the cells was verified at the bottom of the wells, and 400 μl complete medium was added to each well. Cells were resuspended by pipetting. The cell suspensions were transferred to 1.5 ml sterile tubes and centrifuged at 1,500×$g$ for 5 min at room temperature. The supernatant was discarded to remove the transduction reagent, and cells were resuspended in 400 μl fresh complete media and transferred to a new 24-well plate. The infected cells were incubated at 37°C, 5% $CO_2$ incubator for 48 hr. The doxycycline was added to induce the biodegrader expression before analysis by flow cytometry and western blotting.

## Western blotting

Cells were washed with PBS before lysis with RIPA buffer (Sigma-Aldrich) (50 mM Tris-HCl, pH 8.0, 150 mM sodium chloride, 1.0% Igepal CA-630 [NP-40], 0.5% sodium deoxycholate [SDC], and 0.1% sodium dodecyl sulphate) supplemented with Pierce protease inhibitor tablets, EDTA-free (Thermo Fisher). Cells were lysed on ice and vortexed every 5 min for 30 min, then centrifuged at 10,000×$g$, 10 min at 4°C. Protein concentrations were quantified using Pierce BCA Protein assay kit (Thermo Fisher) with the BCA standard curve range from 0 to 2 mg/ml. Equal amounts of protein samples were separated on 10% or 15% SDS-PAGE and subsequently transferred to polyvinylidene fluoride membrane (Amersham, Cytiva) by wet blotting method. The membrane was blocked with 10% non-fat milk (Sigma-Aldrich) in PBST (PBS with 0.1% Tween20 [Sigma-Aldrich]) before incubation with primary antibodies overnight at 4°C on a roller. After the incubation, the membrane was washed with PBST for 5 min and the washing step was repeated five times. The washed membrane was incubated with horseradish peroxidase-conjugated secondary antibodies at room temperature for 1 hr on a roller, then washed with PBST for 5 minutes, repeating the washing step five times. The membrane was incubated with Clarity western blotting substrate (Bio-Rad) for 1 min before exposure to a ChemiDoc Imager (Bio-Rad). For quantification, densitometry analysis of protein expression was analysed using Image Lab software.

## Reverse transcription-PCR

The expression of *LMO1* and *LMO2* was verified using RT-PCR method using the primers shown in *Figure 4—figure supplement 2*. 1×$10^7$ cells were harvested and lysed by using 0.5% NP40 lysis buffer (Thermo Scientific). Total cellular RNA was extracted from the supernatant by using water-saturated phenol pH 6.6 (Invitrogen), 2 M NaAc pH 5.0 was added to 300 mM, and 100% ethanol to 70% to precipitate RNA at –20°C for 2 hr. The pellet was washed with 70% aqueous ethanol and dried at room temperature. The RNA precipitate was dissolved in 10 mM Tris pH 7.5. The final RNA concentration was measured using a NanoDrop spectrophotometer (Thermo Scientific). Two μg of RNA was used for first-strand cDNA synthesis with 100 pmoles oligo(dT)$_{12-18}$ primer (Invitrogen) and 20 units SuperScript

II Reverse Transcriptase (Invitrogen). Primers for LMO1 were forward 5'-GATCCAGCCCAAAGGG AAGCAG-3', reverse 5'-GATAAAGGTGCCATTG AGCTG-3'. Primers for LMO2 were forward 5'-GATT CCTCGGCCATCGAAAGG-3', reverse 5'-GATGTTTGTAGTAGAGGCGCCG-3'. Primers for KRAS were forward 5'-GATATGACTGAATATAAACTTGTGGTAG-3', reverse 5'-GATGGCAAATACACAAAGA AAGC-3'. PCR was performed by using DNA Engine Tetrad Thermal Cycler (Bio-Rad). The PCR conditions were 98°C 1 min 1 cycle for initial denaturation, and 30 cycles of 98°C 10 s for denaturation, 65°C 15 s for annealing, 72°C 30 s per 1 kb for extension, then 72°C 10 min for final extension. The PCR products were resolved on 1.0% agarose gel, visualised by SYBR Safe staining, and quantified using ChemiDoc Imager (Bio-Rad).

## Genomic PCR

The presence of the chromosomal translocation t(11;14)(p13;q11) in KOPT-K1 cells (*Dong et al., 1995*) was confirmed using genomic PCR. The derived genomic translocation sequence is shown in *Figure 4—figure supplement 3*. Total DNA was extracted from $1\times10^6$ KOPT-K1 cells by using DNeasy Blood&Tissue Kit (QIAGEN) according to the manufacturer's instruction. The final DNA concentration was measured using a NanoDrop spectrophotometer (Thermo Scientific). One hundred ng of DNA template was used to generate PCR using Phusion High-Fidelity PCR Kit (NEB). PCR was performed by using DNA Engine Tetrad Thermal Cycler (Bio-Rad). The forward and reverse primers used to confirm the chromosomal translocation in KOPT-K1 were forward two 5'-GATGAAT TCGAAGCTACTG CAGCCATC-3', forward three 5'-GATGAATTCATGCTATGAGGTA GGTATG-3', and J delta reverse 5'-GATGGATCCGGTTCCACAGTCACTCGGGTTCC-3'. The PCR condition was 98°C 1 min 1 cycle for initial denaturation, followed by 98°C 10 s for denaturation, 65°C 15 s for annealing, 72°C 15 s for extension for 30 cycles, then 72°C 10 min for final extension. The PCR products were resolved on 1% agarose gel, visualised by SYBR Safe staining, and quantified using ChemiDoc Imager (Bio-Rad).

## Chemistry: synthesis, purification, and quality control

### Synthesis of Abd-CRBN

Ethyl 2-(4-(3-methoxybenzyl)piperazin-1-yl)oxazole-4-carboxylate

**Chemical structure 1.** ethyl 2-(4-(3-methoxybenzyl)piperazin-1-yl)oxazole-4-carboxylate.

1-(3-Methoxybenzyl)piperazine (150 mg, 0.728 mmol, 1.2 eq.) was dissolved in 1,4-dioxane/*N,N*-diisopropylethylamine (4:1, 5 ml) before addition of ethyl 2-chlorooxazole-4-carboxylate (116 mg, 0.661 mmol, 1.0 eq.). The solution was stirred at 60°C for 48 hr, cooled down to room temperature, diluted with EtOAc (20 ml) and washed with $H_2O$/brine (1:1, 20 ml). The organic phase was dried ($Na_2SO_4$), filtered, and concentrated in vacuo. The crude material was then purified on silica gel (4% MeOH in $CH_2Cl_2$) to afford the title compound as a yellow oil (163 mg, 71%), after purification on silica gel (4% MeOH in $CH_2Cl_2$).

Ethyl 2-(4-(3-methoxybenzyl)piperazin-1-yl)oxazole-4-carboxylate

**Chemical structure 2.** ethyl 2-(4-(3-methoxybenzyl)piperazin-1-yl)oxazole-4-carboxylate.

Ethyl 2-(4-(3-methoxybenzyl)piperazin-1-yl)oxazole-4-carboxylate (200 mg, 0.580 mmol, 1.0 eq.) was dissolved in THF/MeOH (4:1) before addition of NaOH (1 M aq.) until pH>8. The resulting reaction was stirred for 16 hr at room temperature and then acidified with HCl (1 M aq.) until pH<5. The solution was concentrated in vacuo and the obtained carboxylic acid (184 mg, quant.) was used directly in the next step.

## Ethyl 4-(2-(4-(3-methoxybenzyl)piperazin-1-yl)oxazole-4-carboxamido)benzoate

**Chemical structure 3.** ethyl 4-(2-(4-(3-methoxybenzyl)piperazin-1-yl)oxazole-4-carboxamido)benzoate.

2-(4-(3-Methoxybenzyl)piperazin-1-yl)thiazole-4-carboxylic acid (184 mg, 0.580 mmol, 1.0 eq.) was dissolved in DMF (4 ml) before sequential addition of *N,N*-diisopropylethylamine (303 μl, 1.74 mmol, 3.0 eq.), benzocaine (115 mg, 0.696 mmol, 1.2 eq.) and HATU (309 mg, 0.812 mmol, 1.4 eq.). The resulting solution was stirred for 18 hr, diluted with EtOAc (10 ml), and washed with brine/water (1:1, 3×50 ml). The organic phase was dried ($Na_2SO_4$), filtered, and concentrated in vacuo. The title product was obtained as a colourless oil that solidified on standing (247 mg, 92%), after purification on silica gel (4% MeOH in $CH_2Cl_2$).

## 4-(2-(4-(3-Methoxybenzyl)piperazin-1-yl)oxazole-4-carboxamido)benzoic acid

**Chemical structure 4.** 4-(2-(4-(3-methoxybenzyl)piperazin-1-yl)oxazole-4-carboxamido)benzoic acid.

Ethyl 4-(2-(4-(3-methoxybenzyl)piperazin-1-yl)oxazole-4-carboxamido)benzoate (120 mg, 0.252 mmol, 1.0 eq.) was dissolved in THF/MeOH (4:1) before addition of NaOH (1 M aq.) until pH>8. The resulting reaction was stirred for 16 hr at room temperature and then acidified with HCl (1 M aq.) until pH<5. The solution was concentrated in vacuo and the obtained carboxylic acid (114 mg, quant.) was used directly in the next step.

*N*-(4-((6-((2-(2,6-dioxopiperidin-3-yl)-1,3-dioxoisoindolin-4-yl)oxy)hexyl)carba-moyl)phenyl)-2-(4-(3-methoxybenzyl)piperazin-1-yl)oxazole-4-carboxamide

**Chemical structure 5.** N-(4-((6-((2-(2,6-dioxopiperidin-3-yl)-1,3-dioxoisoindolin-4-yl)oxy)hexyl)carbamoyl)phenyl)-2-(4-(3-methoxybenzyl)piperazin-1-yl)oxazole-4- carboxamide.

4-(2-(4-(3-Methoxybenzyl)piperazin-1-yl)oxazole-4-carboxamido)benzoic (65 mg, 0.150 mmol, 1.2 eq.) was dissolved in DMF (1 ml) before sequential addition of *N,N*-diisopropylethylamine (66 μl, 0.381 mmol, 3.0 eq.), 4-((6-aminohexyl)oxy)-2-(2,6-dioxopiperidin-3-yl)isoindoline-1,3-dione (47 mg, 0.127 mmol, 1.0 eq.) and HATU (80 mg, 0.210 mmol, 1.4 eq.). The resulting solution was stirred for 18 hr, diluted with EtOAc (10 ml) and washed with brine/water (1:1, 3×50 ml). The organic phase was dried (Na$_2$SO$_4$), filtered, and concentrated in vacuo. The title product was obtained as a colourless oil (89 mg, 89%), after purification on silica gel (10% MeOH in CH$_2$Cl$_2$).

## Synthesis of Abd-VHL
*tert*-Butyl 2-(2-(2-aminoethoxy)ethoxy)acetate was prepared following literature precedent.

**Chemical structure 6.** Synthesis of Abd-VHL.

*tert*-Butyl 2-(2-(2-(4-(2-(4-(3-methoxybenzyl)piperazin-1-yl)thiazole-4-carbox-amido)benzamido)ethoxy)ethoxy)acetate

**Chemical structure 7.** tert-butyl 2-(2-(2-(4-(2-(4-(3-methoxybenzyl)piperazin-1-yl)thiazole-4-carboxamido) benzamido)ethoxy)ethoxy)acetate.

4-(2-(4-(3-Methoxybenzyl)piperazin-1-yl)thiazole-4-carboxamido)benzoic (36 mg, 0.080 mmol, 1.0 eq.) was dissolved in DMF (1 ml) before sequential addition of *N,N*-diisopropylethylamine (42 μl, 0.240 mmol, 3.0 eq.), *tert*-butyl 2-(2-(2-aminoethoxy)ethoxy)acetate (20 mg, 0.086 mmol, 1.2 eq.) and HATU (43 mg, 0.112 mmol, 1.4 eq.). The resulting solution was stirred for 18 hr, diluted with EtOAc (10 ml), and washed with brine/water (1:1, 3×50 ml). The organic phase was dried ($Na_2SO_4$), filtered, and concentrated in vacuo. The title product was obtained as a colourless oil (30 mg, 57%), after purification on silica gel (10% MeOH in $CH_2Cl_2$).

## 2-(2-(2-(4-(2-(4-(3-Methoxybenzyl)piperazin-1-yl)thiazole-4 carboxamido) benzamido) ethoxy)ethoxy) acetic acid

**Chemical structure 8.** 2-(2-(2-(4-(2-(4-(3-methoxybenzyl)piperazin-1-yl)thiazole-4 carboxamido)benzamido) ethoxy) ethoxy) acetic acid.

*tert*-Butyl 2-(2-(2-(4-(2-(4-(3-methoxybenzyl)piperazin-1-yl)thiazole-4-carboxamido)benzamido) ethoxy)ethoxy)acetate (30 mg, 0.046 mmol, 1.0 eq.) was dissolved in $CH_2Cl_2$ (1 ml) before addition of TFA (10 μl) in one portion. The reaction was stirred for 16 hr at room temperature, concentrated in vacuo, and the title product was obtained as a yellow oil (27 mg, quant.) and used directly in the next step.

## VHL ligand

The VHL ligand ((2S,4R)-1-((R)-2-amino-3,3-dimethylbutanoyl)-4-hydroxy-*N*-(4-(4-methylthiazol-5-yl) benzyl)pyrrolidine-2-carboxamide) was prepared following literature precedent with a small modification; the first two steps were replaced by a Suzuki reaction. The final product was columned before use (not carried out in the literature).

**Chemical structure 9.** VHL ligand.

*N*-(4-((2-(2-(2-(((R)-1-((2S,4R)-4-hydroxy-2-((4-(4-methylthiazol-5-yl)benzyl) carbamoyl) pyrrolidine-1-yl)-3,3-dimethyl-1-oxobutan-2-yl)amino)-2-oxoe-thoxy)ethoxy)ethyl)carbamoyl)phenyl)-2-(4-(3-methoxybenzyl)piperazin-1-yl) thiazole-4-carboxamide

**Chemical structure 10.** N-(4-((2-(2-(2-(((R)-1-((2S,4R)-4-hydroxy-2-((4-(4-methylthiazol-5-yl)benzyl)carbamoyl) pyrrolidine-1-yl)-3,3-dimethyl-1-oxobutan-2-yl)amino)-2 oxoethoxy)ethoxy)ethyl) carbamoyl)phenyl)-2-(4-(3-methoxybenzyl)piperazin-1-yl)thiazole-4-carboxamide.

2-(2-(2-(4-(2-(4-(3-Methoxybenzyl)piperazin-1-yl)thiazole-4 carboxamido)benzamido) ethoxy) ethoxy)acetic acid (20 mg, 0.033 mmol, 1.0 eq.) was dissolved in DMF (1 ml) before sequential addition of *N*,*N*-diisopropylethylamine (17 µl, 0.100 mmol, 3.0 eq.), 2S,4(R)-1-((R)–2-amino-3,3-dimethylbutanoyl)-4-hydroxy-*N*-(4-(4-methylthiazol-5-yl)benzyl)pyrrolidine-2-carboxamide (17 mg, 0.040 mmol, 1.2 eq.) and HATU (18 mg, 0.046 mmol, 1.4 eq.). The resulting solution was stirred for 18 hr, diluted with EtOAc (10 ml) and washed with brine/water (1:1, 3×50 ml). The organic phase was dried (Na$_2$SO$_4$), filtered, and concentrated in vacuo. The title product was obtained as a colourless oil (9 mg, 27%), after purification on silica gel (18% MeOH in CH$_2$Cl$_2$), followed by two successive preparative TLC (15% MeOH in CH$_2$Cl$_2$ with double elution).

## Cell assays with Abd-CRBN, Abd-VHL, and inhibitors

Assays for the LMO2 PROTAC compounds Abd-CRBN and Abd-VHL were carried out with compounds synthesised by contract with O2H Discovery. All compounds (Abd compounds and epoxomicin, thalidomide, VHL, and MLN4924) were dissolved in 100% DMSO at 10 mM and stored in small aliquots at –20°C. The stability of the compounds was evaluated before using for experiments by using LC-MS method using a Xevo TQ Mass Spectrometer instrument.

## Cell viability assays

The number of viable cells was measured by staining cells with 0.4% Trypan Blue solution (Invitrogen) and counting with Countess Automated Cell Counter (Invitrogen). For inhibition studies, KOPT-K1 cells were seeded in a six-well plate at 5×10$^5$ cells/well. After the incubation at 37°C in a humidified incubator with 5% CO$_2$ for 24 hr, cells were pre-treated with inhibitor compounds (epoxomicin, thalidomide, or lenalidomide) or Neddylation inhibitor (MLN4924) for 2 hr or proteasome inhibitor (epoxomicin) for 24 hr prior to the treatment of Abd compounds for 24 hr. For apoptosis studies, KOPT-K1 cells were seeded in a six-well plate at 5×10$^5$ cells/well. After the incubation at 37°C in a humidified incubator with 5% CO$_2$ for 24 hr, cells were treated with Abd-CRBN or Abd-VHL for 24 or 48 hr.

Cell viability was also assessed by CellTiter-Glo (Promega) according to the manufacturer's instructions. Cells were seeded in triplicate in white 96-well plates (PerkinElmer) at 1×10$^4$ cells/well. After the treatment with compounds, an equal amount of CellTiter-Glo reagent was added to each well and incubated at room temperature for 10 min before the luminescence signal measurement using PHERAstar FSX (BMG Labtech).

## Dose response and EC$_{50}$ determination

KOPT-K1, Jurkat, and MOLT-3 cells were seeded in white 96-well plate at $1\times10^4$ cells/well and incubated at 37°C in a humidified incubator with 5% $CO_2$. After 24 hr incubation, cells were treated with 5, 10, 15, 20, or 25 µM of Abd-CRBN or Abd-VHL for 24 hr. The viable cells were measured using the CellTiter-Glo assay. The EC$_{50}$ was calculated using Prism 9 GraphPad Software. For dose response by western blotting analysis, KOPT-K1 and Jurkat cells were seeded in six-well plates at $5\times10^5$ cells/well and the incubation at 37°C in a humidified incubator with 5% $CO_2$ for 24 hr. Then, cells were treated with Abd-CRBN or Abd-VHL compounds at concentrations of 5, 10, or 20 µM for 2, 6, or 24 hr.

## Caspase3/7 activity assay

Caspase activities were measured using Caspase-Glo 3/7 (Promega) according to the manufacturer's instruction. Cells were seeded in triplicate in white 96-well plates (PerkinElmer) at $1\times10^4$ cells/well. After the treatment with Abd-CRBN or Abd-VHL compounds for 24 hr, an equal amount of CellTiter-Glo reagent was added to each well and mixed using a plate shaker at 300 rpm for 30 s, incubated the plate at room temperature for 1 hr before the luminescence signal measurement using PHERAstar FSX (BMG Labtech).

## Proteomic analysis by mass spectrometry

Samples were prepared using the SimPLIT workflow previously described (*Sialana et al., 2022*) with minor modifications. Cell pellets were lysed in a buffer containing 0.1 M TEAB, 1% SDC, 10% isopropanol, 50 mM NaCl, 5 mM TCEP, 10 mM IAA, nuclease, and protease/phosphatase inhibitors. After 5 min of bath sonication and 45 min at room temperature, protein concentration was measured by Bradford assay. For digestion, 15 µg protein aliquots were treated with trypsin and incubated at 37°C for 2 hr, then overnight at room temperature. Digested samples were acidified, spun, desalted, and labelled with TMTpro-18plex. Labelled samples were combined, acidified, centrifuged to remove SDC, and dried.

Offline peptide fractionation was based on high pH Reverse Phase (RP) chromatography using the Waters XBridge C18 column (2.1×150 mm², 3.5 µm) on a Dionex Ultimate 3000 HPLC as previously described (*Thomas et al., 2024*) with minor modifications. Retention time-based fractions are collected and pooled into 24 samples for LC-MS analysis.

Samples were analysed on a Dionex UltiMate 3000 UHPLC with an Orbitrap Ascend Tribrid mass spectrometer and a PepMap C18 capillary column, as previously described (*Thomas et al., 2024*). A 120 min gradient separation was applied. Data acquisition used TMT-SPS-MS3 with real-time database search, targeting precursors between 400 and 1600 m/z, and employing RTS spectral identification of MS2 fragments to trigger SPS-MS3 scans.

Targeted proteomics analysis by mass spectrometry was further conducted on the fractions containing LMO1 and LMO2 peptides detected in the global proteomics analysis to improve sensitivity. Precursors (593.01, 582.34, 693.68, 745.72, 888.52, and 921.39 m/z) were selected in the quadrupole with an isolation width of 0.7 m/z and fragmented with HCD using 36% collision energy. MS2 spectra were recorded in profile mode in Orbitrap at 45,000 resolutions using a 20-scan mode for the isolation list and an AGC setting of $5\times10^5$.

Mass spectra were analysed using SEQUEST-HT and COMET in Proteome Discoverer 3.0 for protein identification and quantification. Parameters included a 20 ppm precursor mass tolerance, 0.5 Da for TMT-MS3, and 0.02 Da for targeted MS2. Searches targeted fully tryptic peptides with up to 2 mis-cleavages, with TMTpro and carbamidomethylation as static modifications, and methionine oxidation and asparagine/glutamine deamidation as variable. Peptide confidence was estimated with Percolator, FDR set at 0.01, and validated by q value and decoy database search.

### Proteomics data normalisation, analysis, and visualisation

Each TMT column was normalised by dividing the protein abundance values by the median of each value per column. The normalised data were used to calculate log$_2$ ratios of treated vs DMSO for all cell lines. GSEA (*Subramanian et al., 2005*) was conducted to evaluate the biological effects of 2294 treatment by analysing log$_2$ ratios of protein abundance from global proteomics data. The proteins were ranked based on log$_2$ ratios, indicating up-regulation or down-regulation in 2294-treated samples compared to DMSO controls. Significant GSEA enrichment scores were calculated using the

annotation terms using KEGG and GO annotation terms, applying a p-value threshold of <0.05 with Benjamini-Hochberg adjustment. Significant terms were visualised using ridge plots and a heatmap summarising the top 10 terms based on the adjusted p-value.

## Acknowledgements

This work was financed by grants from Blood Cancer UK (BCUK 12051 and 19013), CR-UK (Project Development Award), and by ICR Core. We wish to thank Prof. Marc Mansour for providing T-ALL cell lines (P12-Ichikawa, MOLT-4, PF-382, DND-41, and ALL-SIL) and very grateful to Teresa Palomero, Foon Wu-Baer, and Richard Baer for LOUCY, SUP-TI, and RPMI8402 cells. We also thank various ICR colleagues for generously providing reagents, the ICR Flow Cytometry Facility for their support, services, and expertise, and Dr John Caldwell for advice on chemical synthesis and O2H Group for batch synthesis of the Abd degrader compounds.

## Additional information

### Competing interests

Angela Russell, Terence Rabbitts: Co-founder and share-holder in Kodiform Therapeutics. The other authors declare that no competing interests exist.

### Funding

| Funder | Grant reference number | Author |
|---|---|---|
| Blood Cancer UK | 12051 | Naphannop Sereesongsaeng Carole Bataille Nicolas Bery Ami Miller Terence Rabbitts |
| Cancer Research UK | CRUK PDF | Naphannop Sereesongsaeng Terence Rabbitts |
| Blood Cancer UK | 19013 | Naphannop Sereesongsaeng Carole Bataille Nicolas Bery Ami Miller Terence Rabbitts |

The funders had no role in study design, data collection and interpretation, or the decision to submit the work for publication.

### Author contributions

Naphannop Sereesongsaeng, Investigation, Methodology, Writing – original draft, Writing – review and editing; Carole Bataille, Investigation, Methodology; Angela Russell, Conceptualization, Supervision, Funding acquisition, Investigation; Nicolas Bery, Fernando J Sialana, Ami Miller, Investigation; Jyoti Choudhary, Methodology; Terence Rabbitts, Conceptualization, Supervision, Funding acquisition, Writing – original draft, Writing – review and editing

### Author ORCIDs

Naphannop Sereesongsaeng ⓘ https://orcid.org/0000-0001-7344-441X
Nicolas Bery ⓘ https://orcid.org/0000-0002-2643-3897
Fernando J Sialana ⓘ https://orcid.org/0000-0001-9083-1657
Terence Rabbitts ⓘ https://orcid.org/0000-0002-4982-2609

Reviewer #2 (Public review): https://doi.org/10.7554/eLife.106699.3.sa1
Author response https://doi.org/10.7554/eLife.106699.3.sa2

## Additional files

### Supplementary files
MDAR checklist

### Data availability
All data generated or analysed during this study are included in the manuscript and supporting files.

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
