## [Editor Report · eLife Assessment]

This **important** paper reports the development of proteins and small molecules that induce degradation of a clinically-relevant oncogenic transcription factor, LMO2. The findings provide a proof of concept that PROTAC-type chemicals can be developed against intrinsically disordered proteins. The methods provide a blueprint for rational design of PROTACs starting from intracellular antibody paratopes. Overall, the paper is supported by **solid** evidence and will be of interest to chemical biologists and cancer pharmacologists.

---

## [Referee Report · Reviewer #2 (Public review)]

Summary:

Sereesongsaeng et al. aimed to develop degraders for LMO2, an intrinsically disordered transcription factor activated by chromosomal translocation in T-ALL. The authors first focused on developing biodegraders, which are fusions of an anti-LMO2 intracellular domain antibody (iDAb) with cereblon. Following demonstrations of degradation and collateral degradation of associated proteins with biodegraders, the authors proceeded to develop PROTACs using antibody paratopes (Abd) that recruit VHL (Abd-VHL) or cereblon (Abd-CRBN). The authors show dose-dependent degradation of LMO2 in LMO2+ T-ALL cell lines, as well as concomitant dose-dependent degradation of associated bHLH proteins in the DNA-binding complex. LMO2 degradation via Abd-VHL was also determined to inhibit proliferation and induce apoptosis in LMO2+ T-ALL cell lines.

Strengths:

The topic of degrader development for intrinsically disordered proteins is of high interest and the authors aimed to tackle a difficult drug target. The authors evaluated methods including the development of biodegraders, as well as PROTACs that recruit two different E3 ligases. The study includes important chemical control experiments, as well as proteomic profiling to evaluate selectivity.

Weaknesses:

Several weaknesses remain in this study:

(1) The overall degradation achieved is not highly potent (although important proof-of-concept);

(2) The mechanism of collateral degradation is not completely addressed. The authors acknowledge possible explanations, which would require mutagenesis and structural studies to further dissect;

(3) The proteomics experiments do not detect LMO2, which the authors attribute to its size, making it difficult to interpret.

---

## [Author Response]

The following is the authors’ response to the original reviews.

**Reviewer #1 (Public review):**
Summary:The authors describe the degradation of an intrinsically disordered transcription factor (LMO2) via PROTACs (VHL and CRBN) in T-ALL cells. Given the challenges of drugging transcription factors, I find the work solid and a significant scientific contribution to the field.Strengths:(1) Validation of LMO2 degradation by starting with biodegraders, then progressing to chemical degrades.(2)interrogation of the biology and downstream pathways upon LMO2 degradation collateral degradation §(3) Cell line models that are dependent/overexpression of LMO2 vs LMO2 null cell lines.(4) CRBN and VHL-derived PROTACs were synthesized and evaluated.Weaknesses:(1) The conventional method used to characterize PROTACs in the literature is to calculate the DC50 and Dmax of the degraders, I did not find this information in the manuscript.

As noted in the reply to referee’s point 4 below, our first generation compounds are not highly potent. The DC_50_ values have been computed specifically using Western blot reflected in the data shown in Fig. 2. The revised version Supplementary Fig. S3 shows these quantified Western blot data from a time course of treating KOPT-K1 cells with either Abd-CRBN and Abd-VHL, where the 24 hour blot data are shown in Figure 2, G and E, and the quantified data from each 24 hour treatment are quantified in Supplementary Fig. S3. With these data, the DC_50_ values 9 μM for Abd-CRBN and 15 μM Abd-VHL, included in in the main text and the Supplementary Fig. S3 figure legend.

In addition, the loss of signal of the LMO2-Rluc reporter protein from PROTAC treated cells shown in Fig. 2M has been used to calculate a half-point of degradation; although strictly not DC_50_, as it measures a reporter protein, this yielded values are 10 μM for Abd-CRBN and 9 μM Abd-VHL.

(2) The proteomics data is not very convincing, and it is not clear why LMO2 does not show in the volcano plot (were higher concentrations of the PROTAC tested? and why only VHL was tested and not CRBN-based PROTAC?).

Due to the relatively small size of the LMO2 protein, it is challenging to produce enough unique peptides for reliable identification, especially to distinguish some proteins in the LMO2 complex.

(3) The correlation between degradation potency and cell growth is not well-established (compare Figure 4C: P12-Ichikawa blots show great degradation at 24 and 48 hrs, but it is unclear if the cell growth in this cell line is any better than in PF-382 or MOLT-16) - Can the authors comment on the correlation between degradation and cell growth?

In this study (Fig. 4) we did not aim to compare the effect of LMO2 loss on cell growth among LMO2 positive cells. Rather, we aimed to evaluate the LMO2 importance for cell growth in LMO2-expressing T-ALL cells compared to non-expressing cells and to correlate the loss of the protein with this effect on the cell growth. In addition, the treatment of cells with the LMO2 compounds did now show an effect to LMO2 negative cells until at least 48 hours of treatment indicating that low toxicity of our PROTAC compounds and providing correlation between LMO2 loss and cell growth.

(4) The PROTACs are not very potent (double-digit micromolar range?) - can the authors elaborate on any challenges in the optimization of the degradation potency?

The Abd methodology to use intracellular domain antibodies to screen for compounds that bind to intrinsically disordered proteins such as the LMO2 transcription factors offers a tractable approach to hard drug targets but, in so doing, creates challenging factors to improve the potency that are not the same as those targets for which structural data are available. LMO2 is an intrinsically disordered protein, for which soluble recombinant protein is not readily available to identify the binding pocket of compounds. The potency has so far been optimized solely based on the different moieties substituted in cell-based SAR studies (http://advances.sciencemag.org/cgi/content/full/7/15/eabg1950/DC1) and all new compounds were tested with BRET assays. Thus, currently optimization of the degradation potency (including properties such as improved solubility) for the LMO2-binding compounds relies on chemical modification the three areas of the compounds indicated in Fig. 2 B,C.

(5) The authors mentioned trying six iDAb-E3 ligase proteins; I would recommend listing the E3 ligases tried and commenting on the results in the main text.

The six chimaeric iDAb-E3 ligase proteins involved one anti-LMO2 iDAb and three different E3 ligase where either fused at the N- or the C-terminus of the VH (giving six protein formats). These six fusion proteins were described in the text referring to the degrader studies described in Supplementary Fig. 1.

**Reviewer #2 (Public review):**
Summary:Sereesongsaeng et al. aimed to develop degraders for LMO2, an intrinsically disordered transcription factor activated by chromosomal translocation in T-ALL. The authors first focused on developing biodegraders, which are fusions of an anti-LMO2 intracellular domain antibody (iDAb) with cereblon. Following demonstrations of degradation and collateral degradation of associated proteins with biodegraders, the authors proceeded to develop PROTACs using antibody paratopes (Abd) that recruit VHL (Abd-VHL) or cereblon (Abd-CRBN). The authors show dose-dependent degradation of LMO2 in LMO2+ T-ALL cell lines, as well as concomitant dose-dependent degradation of associated bHLH proteins in the DNA-binding complex. LMO2 degradation via Abd-VHL was also determined to inhibit proliferation and induce apoptosis in LMO2+ T-ALL cell lines.Strengths:The topic of degrader development for intrinsically disordered proteins is of high interest, and the authors aimed to tackle a difficult drug target. The authors evaluated methods, including the development of biodegraders, as well as PROTACs that recruit two different E3 ligases. The study includes important chemical control experiments, as well as proteomic profiling to evaluate selectivity.Weaknesses:The overall degradation is relatively weak, and the mechanism of potential collateral degradation is not thoroughly evaluated

The purpose of the study was to evaluate effects of LMO2 degraders. The mechanism of the observed collateral degradation could not be investigated directly within the scope of our study. In the main text, discussed two possible, not exclusive, explanations. One being that our work (and previously published, cited work) indicates that the DNA-binding bHLH proteins have relatively short half file (Supplementary Fig. S12) and may therefore be subject to normal turnover when the LMO2, which is in the complex, turns over. Further, the known structure of the LMO2-bHLH interactions (from Omari et al, doi: 10.1016/j.celrep.2013.06.008) was also examined for the location of lysines in the TAL1 & E47 partners (Supplementary Fig. S11). It is possible that their local association with the LMO2-E3-ligase complex created by the PROTAC interaction, could cause their concurrent degradation. Mutagenesis and structural analysis would be needed to establish this point.

In addition, experiments comparing the authors' prior work with their anti-LMO2 iDAb or Abl-L are lacking, which would improve our understanding of the potential advantages of a degrader strategy for LMO2.

A major motivation behind developing the Antibody-derived (Abd) method to select compounds, which are surrogates of the antibody paratope, is because using iDAbs directly as inhibitors requires the development of delivery technologies for these macromolecules, as protein directly or as vectors or mRNA for their expression. Ultimately, high affinity anti-LMO2 iDAbs should directly be used as tractable inhibitors when delivery methods redeveloped. In the meantime, Abd compounds were envisaged as being surrogates suitable for development into reagents, and potentially drugs, by medicinal chemistry. We evaluated selected first generation LMO2-binding Abd compounds previously, finding their ability to interfere with LMO2-iDAb BRET signal to EC_max_ about 50% but these compounds do not have potency to have an effect on the interaction of LMO2 with a non-mutated iDAb (nM affinity). These data indicated that efficacy improvement for the PROTACs was needed. In addition, in the current study, we observed viability effects in T-ALL lines at high concentrations (20 μM) irrespective of LMO2 expression (Supplementary Fig. S 2A, B) These data indicated that efficacy improvement was needed and potentially converting the degraders (PROTACs) would add to in-cell potency. By adding the E3 ligase ligands, we found the toxicity of non-LMO2 expressing Jurkat was significantly reduced (Supplementary Fig. S 2E, F).

**Reviewer #2 (Recommendations for the authors):**
Suggestions for additional experiments:(1) The data presented is primarily focused on demonstrating targeted degradation of LMO2, with a focus on phenotypes such as proliferation and apoptosis. In this manuscript, there are limited comparative evaluations of anti-LMO2 iDAb or Abl-L to show the potential benefits of a degrader approach to their previously described work, as well as why targeted degradation is in fact, advantageous. For example, the authors' previous work has shown that anti-LMO2 iDAb inhibits tumor growth in a mouse transplantation model. Comparisons in vitro would be supportive of the importance of continued degrader optimization/development.

we have previously shown that an anti-LMO2 scFv inhibits tumour growth in a mouse model but this work used an expressed scFv antibody that binds to LMO2 in nM range. The Abd compounds are much lower potency that the antibody and, because recombinant LMO2 is difficult to work with, we could only evaluate interactions of compounds with LMO2 in cell-based assays like BRET (LMO2-iDAb BRET). In this cell-based assay, the first generation Abd compounds do not have sufficient potency to block LMO2-iDAb interaction unless the affinity of the iDAb is reduced to sub-μM. The justification for proceeding on the degrader process rather than just using the protein-protein interaction (PPI) inhibition was based largely around the low potency of the first generation PPI compounds in cell assays and that incorporation protein degradation with PPI inhibition would enhance the efficacy.

In addition, the viability experiments are also very short-term; is there a reason why the authors did not carry out these experiments for 3-5 days to fully understand the impacts on proliferation?

In Supplementary Fig. S5, we did show assays up to 3 days. In KOPT-K1 (LMO2+), the LMO2 levels were reduced during the time course of this assay (from a single compound dose at time zero) (Supplementary Fig S 5A, B). We also show CellTitreGlo assays up to 3 days and, with these second generation compounds, we observed sustained effects on KOPT-K1 (LMO2+) but low non-DMSO toxicity in Jurkat (LMO2-) revised version Supplementary (Fig S5 C, D).

(2) The potential mechanism of collateral degradation is interesting and important in evaluating the on-target responses and consequences of degrading LMO2. At this time, the data supporting collateral degradation is limited and would be strengthened by showing that it is not due to a change in mRNA levels and not due to complex dissociation. Overall, the kinetics and depth of loss of complex members such as E47 in Figure 3 appear more substantial than LMO2 itself, and as presented, collateral degradation is not effectively demonstrated. In addition, to aid in the readers' assessments, additional background and references around the roles of TAL1 and E47 would be helpful. For example, structurally, where do they (and other associated proteins that are not degraded) fit in the complex?

We have responded above in relation to the Public Review Comments and note that a structure of the complex was in submitted version (now revised version Supplementary Fig. S11).

(3) In Figure 1A, the blots show decreased levels of endogenous CRBN with iDAB-CRBN. Is this a known consequence of this approach in these cell lines? Does the partial recovery of endogenous CRBN in KOPTK1 cells have any indication of iDAB-CRBN levels?

We cannot be sure why the endogenous level of CRBN decreases in doxycycline treated cells. It has been shown (DOI:10.1371/journal.pone.0064561) that doxycycline used in the inducible expression system (and its derivatives), such as the lentivirus we used, has an effect to gene expression patterns, which can be increase or decrease expression. Although the published study did not examine CRBN expression, the effect might explain the CRBN expression decrease on doxycycline addition and remains the same level after that.

(4) In Figure S7, the authors do not fully explain the results and why there is minimal rescue with epoxomicin (S7A) or MLN4924 (S7J). This could indicate an alternative mechanism of degradation and loss at play, given the lack of rescue. Can the authors comment on this discrepancy, and have they looked autophagy inhibitor or other agents to achieve the chemical rescue?

In the experiments such as in revised version Supplementary Fig. S6, we used KOPT-K1 cells with a single concentration of the inhibitors and the cells may less susceptible to the epoxomicin (0.8 μM) but lenalidomide and free thalidomide restored the LMO2 levels fully. In the main text Fig. 3D, we also showed that including epoxomicin and thalidomide with the Abd-CRBN in KOPT-K1 and CCRF-CEM restore LMO2 levels, supporting the conclusion that the main mechanism of degradation is through ubiquitination proteosomal route.

(5) For the proteomics data, it would be helpful to have the proteins in yellow highlighted to have them noted in 5D and 5E. In addition, can the authors comment on why LMO2 or their collateral targets are not confirmed in the table? Furthermore, 5C is difficult to interpret; if there are no significantly changing proteins in the Jurkat cells, why are there pathways that are identified?

As mentioned in reply to referee 1, due to the relatively small size of the LMO2 protein, it is challenging to produce enough unique peptides for reliable identification, especially to distinguish some proteins in the LMO2 complex where expression levels are low.